# Single-cell RNA sequencing demonstrates the molecular and cellular reprogramming of metastatic lung adenocarcinoma

Nayoung Kim [1,2,3,13], Hong Kwan Kim[4,13], Kyungjong Lee [5,13], Yourae Hong [1,6], Jong Ho Cho[4], Jung Won Choi[7], Jung-Il Lee[7], Yeon-Lim Suh[8], Bo Mi Ku[9], Hye Hyeon Eum [1,2,3], Soyean Choi [1], Yoon-La Choi[6,10,11], Je-Gun Joung[1], Woong-Yang Park [1,2,6], Hyun Ae Jung[12], Jong-Mu Sun[12], Se-Hoon Lee[12], Jin Seok Ahn[12], Keunchil Park[12], Myung-Ju Ahn [12✉] & Hae-Ock Lee [1,2,3,6✉]

Advanced metastatic cancer poses utmost clinical challenges and may present molecular and cellular features distinct from an early-stage cancer. Herein, we present single-cell transcriptome profiling of metastatic lung adenocarcinoma, the most prevalent histological lung cancer type diagnosed at stage IV in over 40% of all cases. From 208,506 cells populating the normal tissues or early to metastatic stage cancer in 44 patients, we identify a cancer cell subtype deviating from the normal differentiation trajectory and dominating the metastatic stage. In all stages, the stromal and immune cell dynamics reveal ontological and functional changes that create a pro-tumoral and immunosuppressive microenvironment. Normal resident myeloid cell populations are gradually replaced with monocyte-derived macrophages and dendritic cells, along with T-cell exhaustion. This extensive single-cell analysis enhances our understanding of molecular and cellular dynamics in metastatic lung cancer and reveals potential diagnostic and therapeutic targets in cancer-microenvironment interactions.

[1] Samsung Genome Institute, Samsung Medical Center, Seoul 06351, Korea. [2] Department of Molecular Cell Biology, Sungkyunkwan University School of Medicine, Suwon 16419, Korea. [3] Department of Biomedicine and Health Sciences, Graduate School, The Catholic University of Korea, Seoul 06591, Korea. [4] Department of Thoracic and Cardiovascular Surgery, Samsung Medical Center, Sungkyunkwan University School of Medicine, Seoul 06351, Korea. [5] Division of Pulmonary and Critical Care Medicine, Department of Medicine, Samsung Medical Center, Sungkyunkwan University School of Medicine, 06351 Seoul, Korea. [6] Department of Health Sciences and Technology, Samsung Advanced Institute for Health Sciences &Technology, Sungkyunkwan University, Seoul 06355, Korea. [7] Department of Neurosurgery, Samsung Medical Center, Sungkyunkwan University School of Medicine, Seoul 06351, Korea. [8] Department of Pathology, Samsung Medical Center, Sungkyunkwan University School of Medicine, Seoul 06351, Korea. [9] Samsung Biomedical Research Institute, Samsung Medical Center, Sungkyunkwan University School of Medicine, Seoul 06351, Korea. [10] Laboratory of Cancer Genomics and Molecular Pathology, Samsung Medical Center, Sungkyunkwan University School of Medicine, Seoul 06351, Korea. [11] Department of Pathology and Translational Genomics, Samsung Medical Center, Sungkyunkwan University School of Medicine, Seoul 06351, Korea. [12] Division of Haematology-Oncology, Department of Medicine, Samsung Medical Center, Sungkyunkwan University School of Medicine, Seoul 06351, Korea. [13] These authors contributed equally: Nayoung Kim, Hong Kwan Kim, Kyungjong Lee. ✉email: silk.ahn@samsung.com; haeocklee@catholic.ac.kr

Non-small cell lung cancer (NSCLC) is histologically divided into adenocarcinoma, squamous cell carcinoma, and large-cell carcinoma. Lung adenocarcinoma (LUAD) is the most common type, accounting for approximately 40% of all lung cancers. LUAD is often detected at the metastatic stage with prevalence in the brain, bones, and respiratory system[1]. Distant metastasis is the major cause of mortality in lung cancer; however, specific aspects of metastatic lung cancer and its associated microenvironments remain poorly understood.

Efforts made for the understanding of lung cancer progression and metastasis have largely focused on profiling of cancer cells with genetic aberrations[2,3]. However, progression and metastasis are also influenced by complex and dynamic features in tumor surroundings[4]. For instance, application of immune-checkpoint blockades inhibiting PD-1 (programmed cell death protein 1) and CTLA-4 (cytotoxic T-lymphocyte-associated protein 4) in immune cells has opened a new therapeutic window for metastatic NSCLC treatment[5]. The parsing of unique classes of tumor microenvironments in advanced cancer can reveal the key elements involved in the predisposition to tumor-induced immunological changes, and these elements can be exploited for novel immunotherapeutic strategies[6].

Single-cell RNA-sequencing (scRNA-seq) has been recently used for the profiling of tumor microenvironments[7,8]. This technology allows massively parallel characterization of thousands of cells at the transcriptome level. Previous scRNA-seq studies related to lung cancer have been limited to early stage primary tumors and normal tissues resected from a small number of samples of mixed histological types[9,10]. In the present study, we report the comprehensive single-cell transcriptome profiling of LUAD from early to advanced stages of primary cancer and distant metastases, and unveil cellular dynamics and molecular features associated with the tumor progression.

## Results

**Cellular dynamics in early, advanced, and metastatic LUAD.** To elucidate the cellular dynamics in LUAD progression, tumor from primary lung tissues, pleural fluids, and lymph node or brain metastases were obtained from 44 patients with treatment-naïve LUAD during endobronchial ultrasound/bronchoscopy biopsy or surgical resection (Fig. 1a, Supplementary Data 1). Distant normal tissues or lymph nodes were also collected for comparative analyses. We cataloged 208,506 cells into nine distinct cell lineages annotated with canonical marker gene expression (Fig. 1b-d, Supplementary Fig. 1, Supplementary Data 2), thus identifying epithelial (alveolar and cancer cells), stromal (fibroblasts and endothelial cells), and immune cells (T, NK, B, myeloid, and MAST cells) as the common cell types, and oligodendrocytes only in brain metastases (mBrain). Due to the bias introduced during tissue dissociations[9], single-cell RNA sequencing data overestimated the immune cell proportions in comparison to the stromal and epithelial cell types (Supplementary Fig. 2a). In addition, the recovery rate of tumor cells was affected by the histological types of LUAD (Supplementary Fig. 2b). Therefore, we assessed the compositions of immune cell subsets after removing the epithelial and stromal populations. The results faithfully reproduced the immune cell profiles detected by mass cytometry by time of-flight (CyTOF) in early LUAD[11]. The most abundant immune cells at primary tumor sites were observed to be T lymphocytes and myeloid cells (Supplementary Fig. 2c, d). Moreover, we confirmed T and B lymphocyte enrichment and the decline of natural killer (NK) and myeloid cells in early- and advanced-stage lung cancers (tLung and tL/B, respectively) compared to the normal lung tissues (nLung), indicating the activation of adaptive immune responses.

Notably, metastatic lymph nodes (mLN) harbored a significant number of myeloid cells unlike normal lymph nodes (nLN), indicating an association of myeloid infiltration with metastasis. mBrain samples contained immune cells (T, B, and NK cells) at detectable levels as well as resident oligodendrocytes and myeloid cells (microglia). These cellular compositions demonstrated differences in tissue-specific resident populations, as well as gross alterations inflicted by tumor growth and invasion. Thus, our LUAD atlas, revealed cellular dynamics and progression-associated changes in each cellular component at an unprecedented scale and depth.

**Tumor intrinsic signatures associated with LUAD progression.** In the present study, we have explored intrinsic characteristics of adenocarcinoma cells through comparative analysis between normal epithelial and tumor cells from surgical resection. Normal epithelial cells mainly comprised four distinct subpopulations, including alveolar types I (AT1) and II (AT2), and club cells and ciliated cells, expressing well-defined epithelial markers (Supplementary Fig. 3a-c). AT1 and AT2, the most abundant types, can initiate LUAD in the distal airway[12]. In tumor tissues, epithelial cell types may contain residual non-malignant cells along with malignant tumor cells. To separate the definitive tumor cells from a potential non-malignant population, we have used genetic aberrations by inferring copy number variations (CNVs) from the gene expression data[8,13]. The inferred CNV patterns confirmed patient-specific perturbations in malignant tLung, tL/B, mLN, and mBrain cells (Supplementary Fig. 3d). In the subsequent tumor cell analysis, we excluded epithelial cells without CNV present in tumor tissues, because of their ambiguous identity.

Using the definitive tumor and normal epithelial cells, we constructed a transcriptional trajectory[14] (Fig. 2a) to adjust the inter-patient genomic heterogeneity, and find key gene expression programs governing the tumor progression. Indeed, transcriptional states in the trajectory revealed normal differentiation paths as well as progression-associated changes in tumors. Firstly, ciliated epithelial and alveolar cells were located in separate trajectory branches, marking their distinct differentiation states. Secondly, club cells were located between the ciliated and alveolar branches, indicating the intermediate differentiation state[15]. Lastly, tumor cells formed a branched structure, with two transcriptional states (tS1 and tS3) along the normal epithelial cells; however, one (tS2) was observed to be distinctly positioned at the opposite ends of the tS1 and tS3 branches (Fig. 2a, b). In the individual patient-by-patient trajectories, the separation of tS2 from the normal epithelial cells was repeatedly observed (Supplementary Fig. 3e) despite the different trajectory structure in each patient due to the limited representation of cellular components. To identify transcriptional signatures defining cellular states in the trajectory, we selected differentially expressed genes specific to each tumor or normal cell state. Sets of 19, 28, 79, 56, 33, and 248 genes were identified as significantly upregulated signatures in tumor cell states 1, 2, 3 (tS1, tS2, tS3) or normal cell states 1, 2, and 3 (nS1, nS2, nS3) (Supplementary Data 3). Most of S1- and S3-associated genes were shared but differentially regulated between tumor and normal cells, and related to normal epithelial functions maintaining the surfactant homeostasis, lung alveolus development, and cilium movement (Fig. 2c). By contrast, S2-associated genes showed definitive tumor-oriented characteristics, such as aggressive cell movement and abnormal proliferation or apoptosis. Hence, tS1 and tS3 states represented a de-regulation of the normal differentiation programs, whereas the tS2 tumor cell state deviated completely from the normal transcriptional programs.

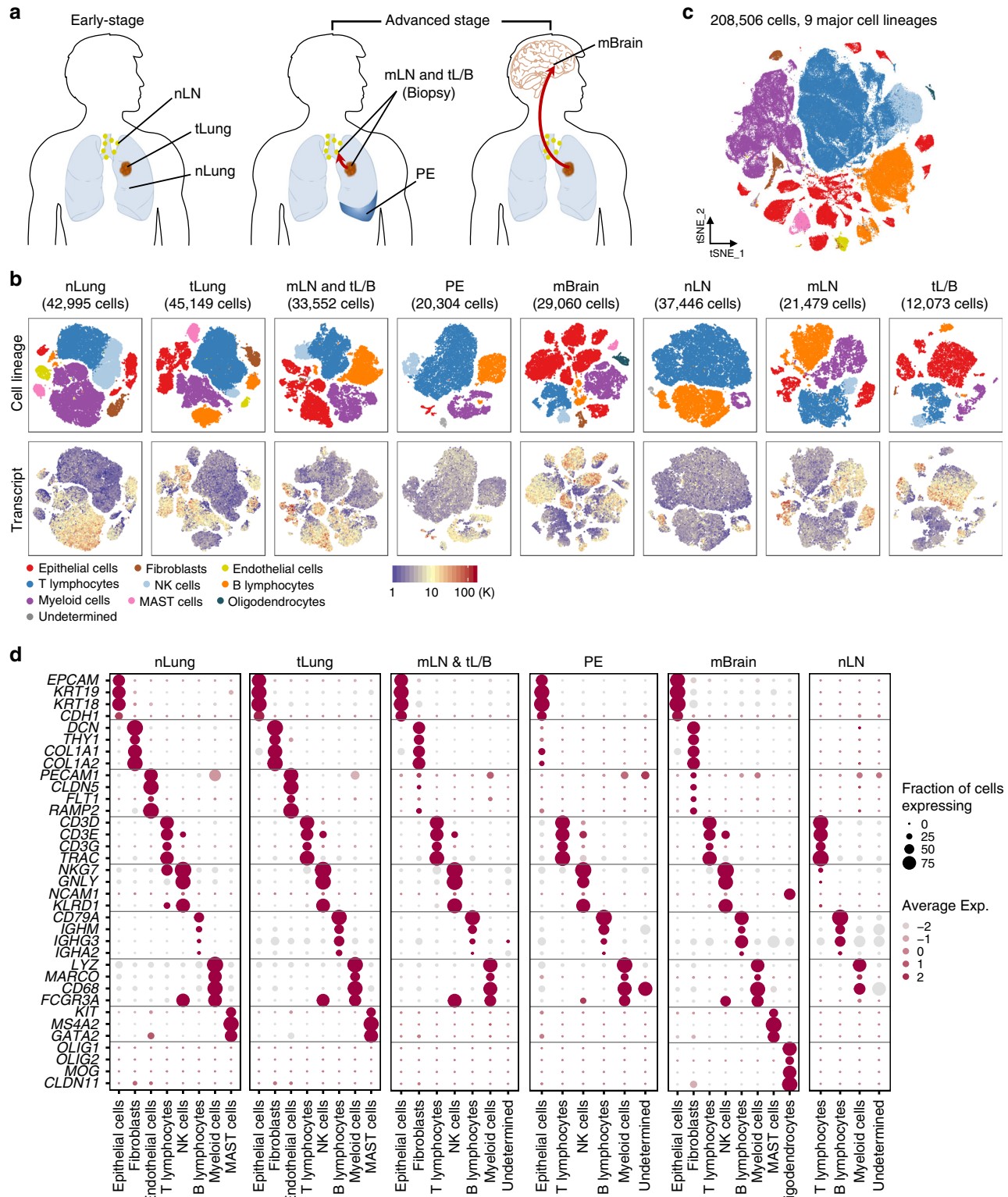

**Fig. 1 Comprehensive dissection and clustering of 208,506 single cells from LUAD patients. a** Overview of tissue origins in the present study collection. Single-cell RNA sequencing was applied to cancer tissue-derived whole cells from primary sites (tLung and tL/B), pleural fluids (PE), lymph node (mLN), and brain metastases (mBrain), as well as normal tissues from lungs (nLung) and lymph nodes (nLN). **b** tSNE projection within each tissue origin, color-coded by major cell lineages and transcript counts. **c** tSNE plot of 208,506 single cells colored by the major cell lineages as shown in (**b**). **d** Dot plot of mean expression of canonical marker genes for nine major lineages from tissues of each origin, as indicated.

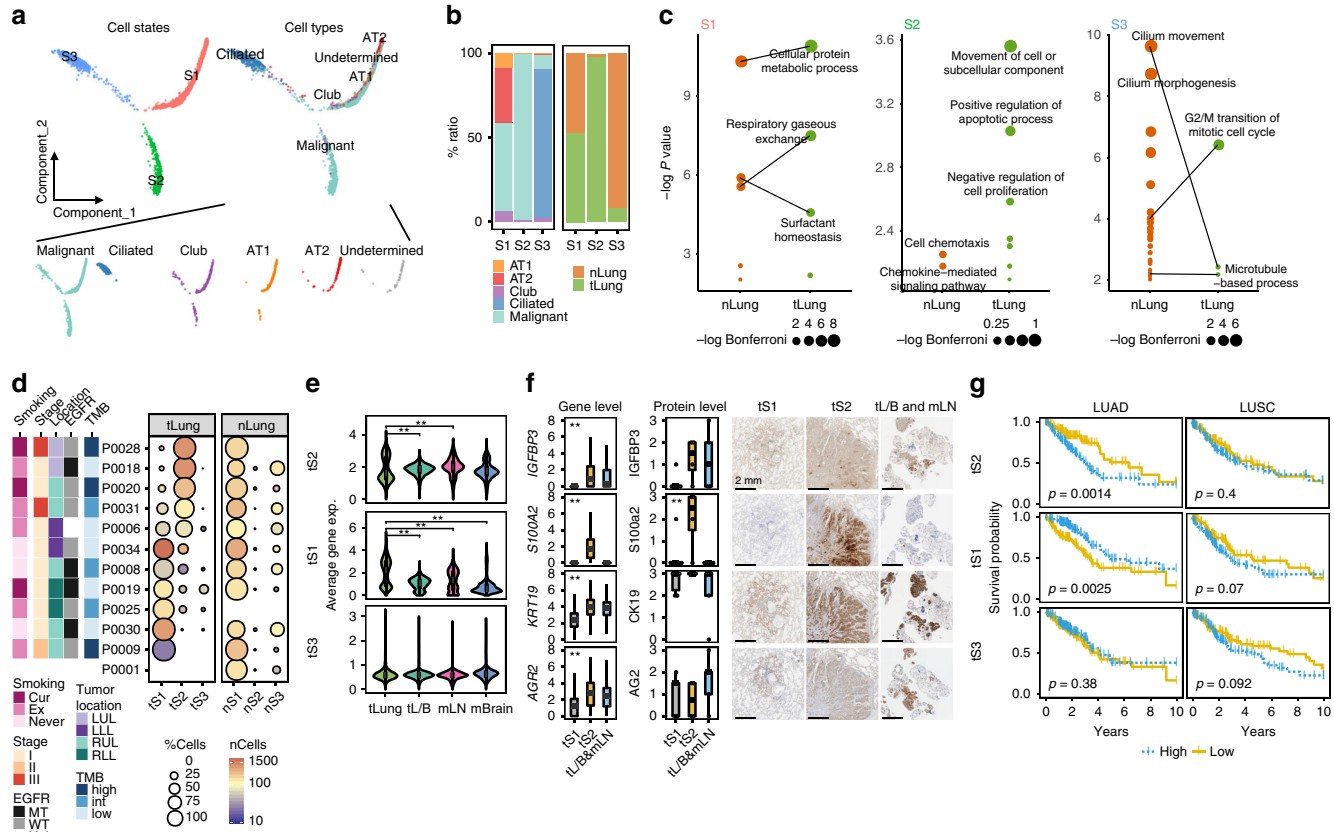

**Fig. 2 Identification of novel cancer cell signature tS2 in LUAD and the connection to poor survival. a** Unsupervised transcriptional trajectory of malignant and normal epithelial cells from Monocle (version 2), colored by cell states and subsets. **b** Relative proportion of cell subsets and tissue origins for each cell state as shown in (**a**). **c** Functional categories (Gene Ontology terms) of signature genes specific to each cell state as shown in (**a**). **d** Different enrichment of cell states within tLung and nLung, and associated clinical parameters. The clinical and pathological parameters originated from tLung. Cur: current smoker; Ex: ex-smoker; Never: never smoker; LUL: left upper lobe; LLL: left lower lobe; RUL: right upper lobe; RLL: right lower lobe. **e** Changes in average tumor cell state-specific signature gene expression with lung cancer progression (tLung and tL/B) and metastasis (mLN and mBrain). **, two-sided Wilcoxon test p-value < 0.01. **f** Box plots depicting single-cell gene expression and the quantified protein levels of selected markers specific to the tS2 epithelial subset. tS1-enriched tLung (n = 7; T06, T08, T09, T19, T25, T30, T34), tS2-enriched tLung (n = 4; T18, T20, T28, T31), and tL/B & mLN (n = 5; EBUS_06, EBUS_10, EBUS_13, EBUS_19, EBUS_28). **, one-way ANOVA test p-value < 0.01. Each box represents the interquartile range (IQR, the range between the 25th and 75th percentile) with the mid-point of the data, whiskers indicate the upper and lower value within 1.5 times the IQR. IHC staining of IGFBP3, S100a2, CK19, and AG2 on formalin-fixed and paraffin-embedded slides for the tS1 (T19), tS2 (T28), and tL/B & mLN (EBUS_10) samples. Scale bar, 2 mm. **g**, Kaplan–Meier overall survival curves of TCGA LUAD (n = 494 samples), and LUSC (n = 490 samples) patients. +: censored observations. P-value (p) was calculated using the two-sided log-rank test.

LUAD patients contained both tS1 and tS2 tumor subpopulations at different fractions, with minor numbers of tS3 (Fig. 2d, tLung). The tS2-specific gene expression was increased for tumor cells that were isolated from the late-stage biopsies or metastases (tL/B, mLN, and mBrain), suggesting an association with tumor progression and metastasis (Fig. 2e). An increase in tS2-specific gene expression was supported at the protein level through immunohistochemical staining of LUAD samples (Fig. 2f). We further tested the clinical impact of the tS2 signature using an independent LUAD cohort from the Cancer Genome Atlas (TCGA). Patients with high tS2 signature gene expression showed worse overall survival (two-sided log-rank test p < 0.01) than those with low expression (Fig. 2g). By contrast, there was no survival difference for lung squamous cell carcinoma (LUSC), indicating an explicit involvement of the tS2 signature with LUAD progression.

To further identify genes related to LUAD progression and/or metastasis, we directly compared tumor cells in early- versus advanced-stage primary, or primary versus metastasis samples (Supplementary Fig. 4, Supplementary Data 4). These pairwise comparisons revealed the gene sets to be differentially regulated during tumor progression and/or metastasis. Survival analysis using these gene sets revealed that late-stage specific gene sets have the highest prediction power for poor survival in LUAD patients.

**Stromal cells orchestrate tissue remodeling and angiogenesis.** To investigate stromal cell dynamics in the tumor microenvironment, we obtained 6168 presumed fibroblasts and endothelial cells as shown in Fig. 1b, and performed a principal component analysis (Supplementary Fig. 5a). The first principal component was observed to split the cells into 2107 endothelial cells and 3794 fibroblasts, with a concordant expression of representative marker genes (average log-normalized expression > 1).

Sub-clustering of endothelial cells (ECs) revealed eight clusters (Fig. 3a). Most EC clusters were observed to belong to the normal tissues and assigned to known vascular cell types, including tip and stalk-like cells, lymphatic ECs, and endothelial progenitor cells[16,17] (Fig. 3b). By contrast, one distinct cluster was identified as tumor-derived ECs (EC-C1) present in tLung and mBrain samples (Fig. 3c, Supplementary Fig. 5b, c). Tumor ECs

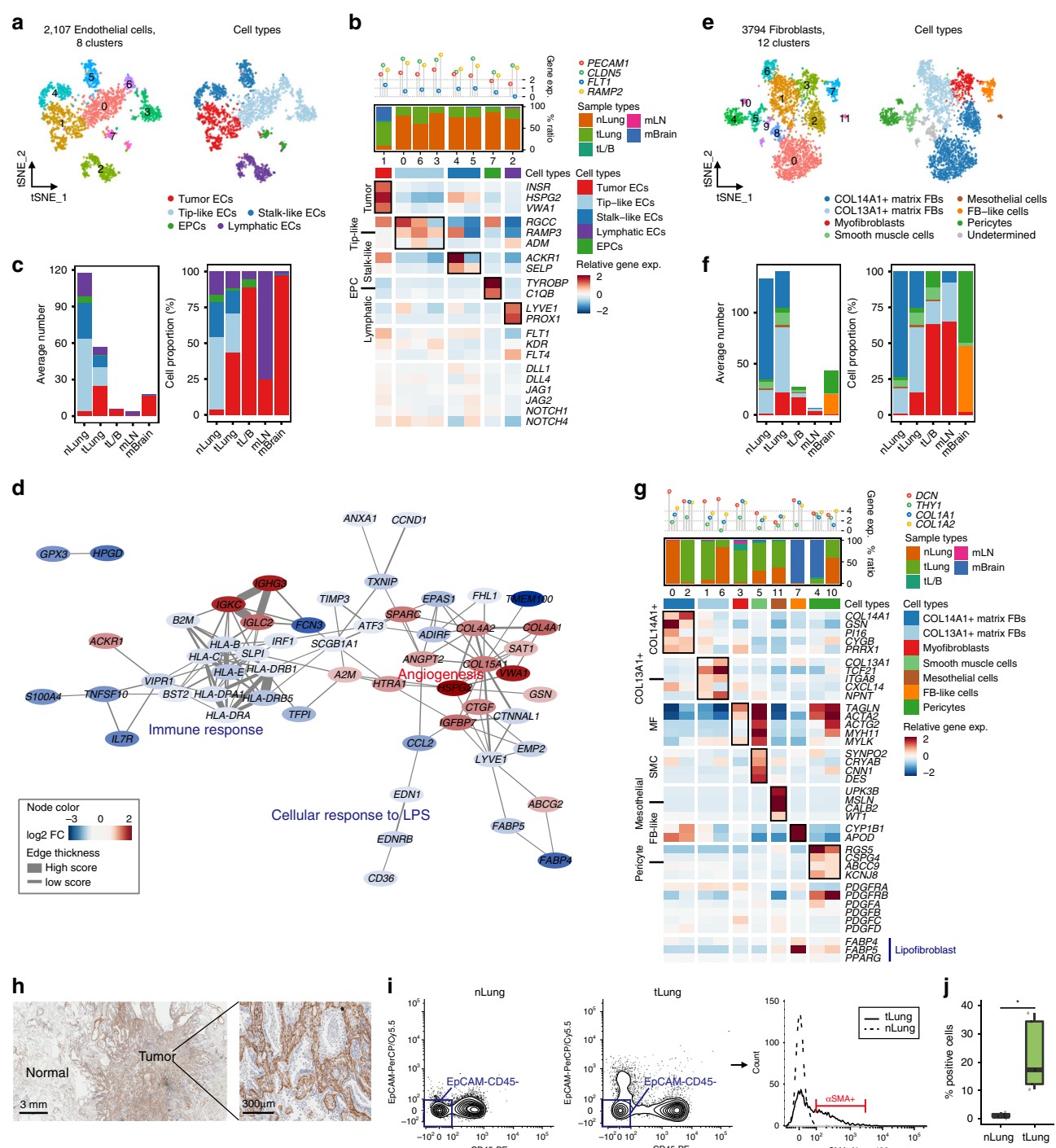

**Fig. 3 Tumor endothelial cells and myofibroblasts promoting angiogenesis and tissue remodeling. a** tSNE plot of endothelial cells, color-coded by clusters and cell subsets as indicated. EPCs: endothelial progenitor cells. **b** Three-layered complex heatmap of selected endothelial cell marker genes in each cell cluster. Top: Mean expression of known lineage markers. Middle: Tissue preference of each cluster; Bottom: Relative expression map of known marker genes associated with each cell subset. Mean expression values are scaled by mean-centering, and transformed to a scale from -2 to 2. **c** Average cell number and relative proportion of EC subsets from tissues of each origin. nLung, $n = 11$ samples; tLung, $n = 11$ samples; tL/B, $n = 3$ samples; mLN, $n = 2$ samples; mBrain, $n = 9$ samples. **d** Functional association networks between signature genes specific to tumor ECs. **e** tSNE plot of fibroblasts (FBs), color-coded by clusters and cell subsets as indicated. **f** Average cell number and relative proportion of fibroblast subsets from tissues of each origin (excluding undetermined cells). nLung, $n = 11$ samples; tLung, $n = 11$ samples; tL/B, $n = 3$ samples; mLN, $n = 4$ samples; mBrain, $n = 10$ samples. **g** Complex heatmap of selected fibroblast marker genes in each cell cluster, as shown in (**b**) (excluding undetermined cells). **h**, IHC staining of α-SMA on formalin-fixed and paraffin-embedded slides for the independent biospecimens ($n = 4$). All replicates showed the similar results. **i**, Representative flow cytometry plots showing myofibroblast (α-SMA+) in primary tumor (T06) and normal lung (N06) tissues. **j**, Box plot of the percentage of myofibroblast (α-SMA+) in normal and tumor-derived EPCAM⁻CD45⁻ cells. nLung ($n = 5$; N06, N18, N41, N42, N43) and tLung ($n = 5$; T06, T18, T41, T42, T43). SMC and pericytes can be included in the α-SMA + cells as minor populations. *$p < 0.05$ ($p = 0.02$), two-sided Student's $t$ test. Each box represents the interquartile range (IQR, the range between the 25th and 75th percentile) with the mid-point of the data, whiskers indicate the upper and lower value within 1.5 times the IQR.

demonstrated a strong activation of VEGF and Notch signaling (Fig. 3b), which regulates the development and cell fate determination of endothelial cells[18,19]. Gene expression network analysis of tumor ECs further highlighted angiogenesis as the upregulated genes' functional category (Fig. 3d, Supplementary Data 5). Thus, brain metastases and primary tumors induced similar vascular changes to accommodate extensive neovascularization. Among the upregulated genes, insulin receptor (INSR) overexpression in the tumor vasculature was suggested as an attractive therapeutic target[20]. Together, these data further supported the therapeutic strategies targeting pro-angiogenic pathways in lung cancer and brain metastases[21,22]. Notably, significantly downregulated genes in tumor ECs were related to immune activation (Fig. 3d, Supplementary Data 5), supporting a previous finding that tumor ECs suppress the immune responses[9,23].

Sub-clustering of fibroblasts revealed 12 distinct clusters, assigned to seven known cell types, including COL13A1+ and COL14A1+ matrix fibroblasts, myofibroblasts, smooth muscle cells, mesothelial cells, fibroblast-like cells in mBrain, and pericytes[24–26] (Fig. 3e). The COL13A1+ and COL14A1+ matrix fibroblasts comprised the main fibroblast types in normal lung (FB-C0 and 6) and early stage tumor (FB-C1 and 2) tissues (Fig. 3f, g; Supplementary Fig. 5d, e). By contrast, myofibroblasts in FB-C3 exclusively originated from tumor tissues, including tLung, tL/B, and mLN samples. Myofibroblasts have been described as cancer-associated fibroblasts promoting extensive tissue remodeling[27], angiogenesis[28], and tumor progression[29]. The myofibroblasts in mLN might be fibroblastic reticular cells, which have been reported to be immunologically specialized myofibroblasts using encapsulated mesenchymal sponges to gather immune cells into the lymph node[30]. The fibroblast-like cells in mBrain (Fig. 3f, g; Supplementary Fig. 5d, e; FB-C7; CYP1B1+ and APOD+)[26] might represent cells within the perivascular space of central nervous system (CNS) that expanded after CNS injury[31]. The infiltration of myofibroblasts in LUAD was confirmed by the expression of the marker protein alpha smooth muscle actin (α-SMA) (ACTA2 gene product) in the tumor stroma (Fig. 3h) and in tumor-derived EPCAM−CD45− cells (Fig. 3i, j; Supplementary Fig. 10). Partial protein expression of α-SMA was observed in the vascular smooth muscle cells in normal tissues. Conclusively, cellular dynamics in endothelial cells and fibroblasts support a consistent phenotypic shift of stromal cells towards promoting tissue remodeling and angiogenesis in LUAD and distant metastases.

**Suppressive immune microenvironment primed by myeloid cells.** Myeloid cells play a critical role in maintaining tissue homeostasis, and regulate inflammation in the lung. Sub-clustering of 42,245 myeloid cells, as shown in Fig. 1b, revealed them to be monocytes, macrophages, and dendritic cells (Fig. 4a, b). Neutrophils were not recovered in our experimental process. Two macrophage types are known to populate the normal adult lung, including the alveolar (AM) type highly expressing the MARCO, FABP4, and MCEMP1 genes, and the interstitial type derived from circulating monocytes[32,33]. Mo-Macs, which are functionally different from tissue-resident macrophages, are recruited and induced to express profibrotic genes during lung fibrosis[34]. We mainly detected the AM type in normal lung tissues, including anti-inflammatory AM (M−C1 and 6; APOE+, CD163+, and C1QB+)[35–37], pro-inflammatory AM (M-C5; IL1B+ and CXCL8+)[38], and actively cycling AM expressing anti-inflammatory markers (M-C13)[39]. By contrast, lung tumor and distant metastasis tissues were strongly enriched in mo-Macs (anti- and pro-inflammatory mo-Macs in M-C0 and 2,

respectively). Both normal and tumor tissues contained clusters of S100A9+ monocytes (M-C3)[39] or dendritic cells (DCs). The remaining clusters displayed origin-specific heterogeneity and diverse macrophage characteristics, including pleural macrophages[40] from pleural fluids (PE) (M-C8 and 9) or microglia and macrophages[41] derived from mBrain samples (M-C11). The pleural macrophages lacked the expression of pro-inflammatory cytokine genes, such as IL1B and CXCL8, but expressed CD163 transcripts, which are associated with a non-inflammatory phenotype. Overall, our data suggest that tumor-associated macrophages (TAMs) in primary lung tumors and distant metastases mainly propagated from mo-Macs that were ontologically different from tissue-resident macrophages (Fig. 4c, Supplementary Fig. 6a, b).

To understand the transcriptional transition from monocytes to TAMs, we performed an unsupervised trajectory analysis to infer changes in the status of macrophages from lung or lymph node samples (Supplementary Fig. 6c, d). Macrophages can manifest diverse functional phenotypes in health and disease conditions, as pro-inflammatory or anti-inflammatory subpopulations[42]. We have detected a serial transformation of pro-inflammatory monocytes into macrophages along the pseudo-time axis, with cells losing their pro-inflammatory nature and gaining anti-inflammatory signatures (Supplementary Fig. 6e, f, Supplementary Data 6). This transition eventually reached a branching point at which the two macrophage subpopulations either retained part of their pro-inflammatory signatures, or were skewed to an anti-inflammatory gene expression phenotype. Normal lung and tumor tissues were enriched in pro- and anti-inflammatory macrophages, respectively. We have also identified anti-inflammatory macrophages in mLN, with an additional population (LN-Mac-S6) expressing the macrophage inflammatory factors (MIF, CXCL3, and CCL20) (Supplementary Fig. 6f). MIF-expressing macrophages also expressed IL1B and TNF at levels comparable to those in pro-inflammatory monocytes, indicating unique macrophage profiles in mLN.

Dendritic cell clusters, as shown in Fig. 4b, manifested a variegated marker gene expression suggesting the presence of heterogeneous DC subpopulations. For a more comprehensive analysis, we re-classified DCs into six subsets, including CD1c+ DCs (Langerhans cells, LCs), CD141+ DCs, CD207+CD1a+ LCs, pDCs (plasmacytoid DCs), CD163+CD14+ DCs, and activated DCs[43,44] (Fig. 4d, e). This refined the minor DC populations within the total myeloid cell clusters (Fig. 4f). Interestingly, pDCs were rarely found in normal lung tissues, but recovered in selected tumor tissues and metastatic lymph nodes (Fig. 4g, h). The pDCs demonstrated an immunosuppressive phenotype[45] represented by the upregulation of leukocyte immunoglobulin-like receptor (LILR) family genes[46], granzyme B (GZMB) production[47], and loss of CD86, CD83, CD80, and LAMP3 activation marker expression[45,48] (Fig. 4i). The presence of pDCs in some LUAD tissues was confirmed through flow cytometry (Fig. 4j, k; Supplementary Fig. 10). Therefore, both mo-Macs and pDCs could create an immunosuppressive microenvironment that possibly caused a sub-optimal tumor antigen presentation in LUAD and distant metastases.

**Activation and perturbation of adaptive immunity.** Tumor-infiltrating B cells have been identified in tertiary lymphoid structures within NSCLC. Mediating an anti-tumor immune response, these cells have been reported to be associated with prolonged patient survival[49]. Relative proportion of B cells was observed to be increased in primary tumors, compared to the nLung samples (Supplementary Fig. 2c, d). In the lymph nodes, B cells were abundantly present regardless of metastases. Sub-

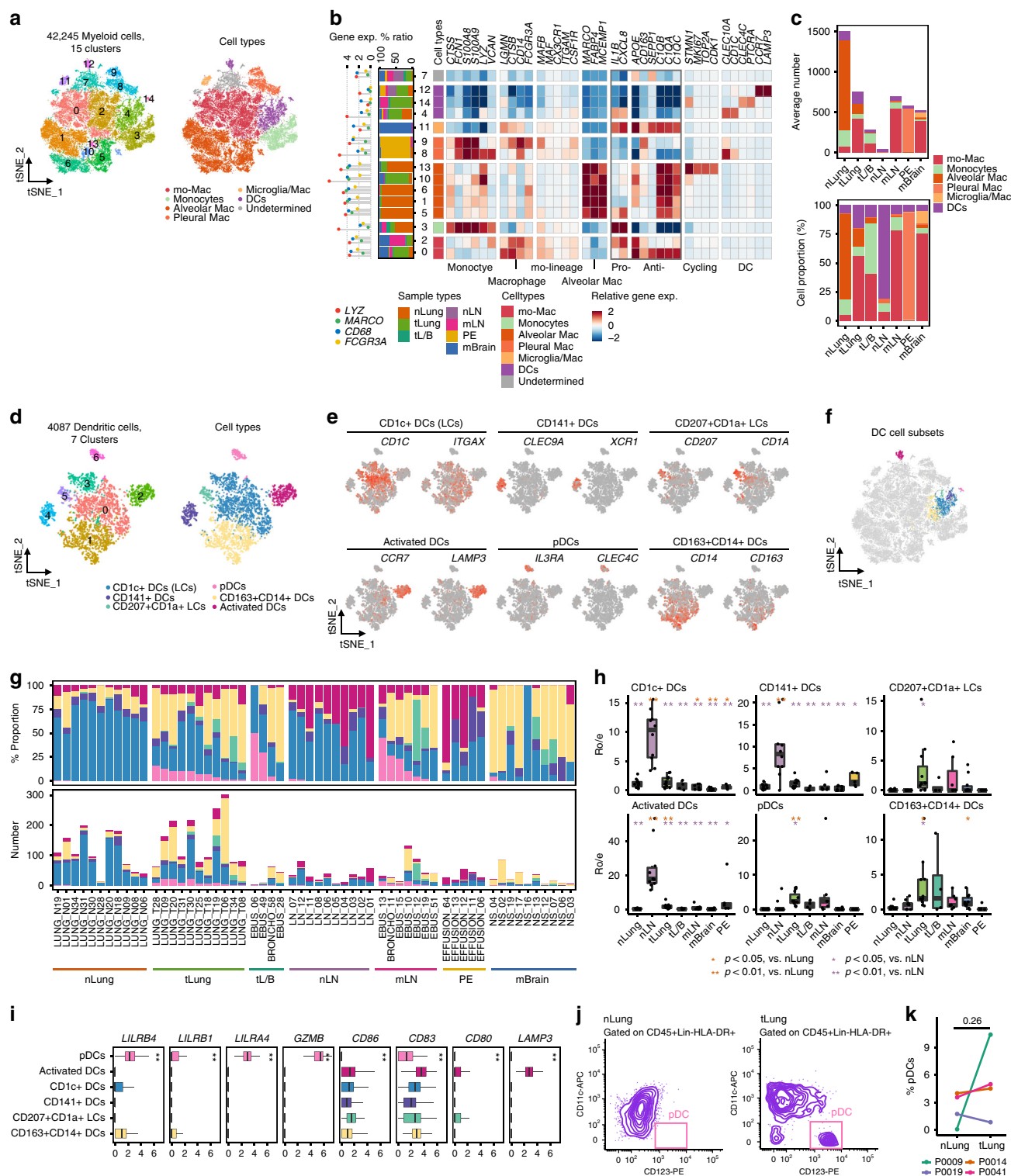

clustering of 27,657 B cells revealed 14 clusters (Fig. 5a) converging to five differentiation states[9,50,51], which were represented as follicular B cells, plasma B cells expressing immunoglobulin gamma (IgG), mucosa-associated lymphoid tissue-derived plasma B cells expressing IgA and joining chain, granzyme B-secreting B cells, and germinal center (GC) B cells (Fig. 5a, Supplementary Fig. 7a, b). Specifically, GC B cells[50] were separated into either dark or light zone cells with distinct transcriptional programs for proliferation or activation, respectively[52] (Supplementary Fig. 7a). Among these, follicular B cells were observed to be the most

abundant in all samples (Fig. 5b, Supplementary Fig. 7c). We have observed tissue-specific enrichment for other subsets (Supplementary Fig. 7c, d). Firstly, normal lung tissues were enriched in granzyme B-secreting cytotoxic cells, whose differentiation was modulated by T-cell-derived IL-12 (ref. [51]). Granzyme B secretion from these cells could play a significant role in mediating cellular cytotoxicity as an alternative to T cells[53]. Secondly, we found more GC B cells in primary tumors and LN metastases than in normal lung and lymph nodes, respectively. These data strongly suggest highly activated humoral immune responses in some

**Fig. 4 Diversity within the myeloid cell lineage and functionality according to tissue origins. a** tSNE plot of myeloid cells, color-coded by clusters and cell subsets as indicated. **b** Complex heatmap of selected myeloid cell marker genes in each cell cluster. Left: Tissue preference of each cluster. Right: Relative expression map of known marker genes associated with each cell subset. Mean expression values are scaled by mean-centering, and transformed to a scale from -2 to 2. Pro-: Pro-inflammatory; Anti-: Anti-inflammatory. **c** Average cell number and relative proportion of myeloid cell subsets from each tissue origin (excluding undetermined cells). nLung, $n = 11$ samples; tLung, $n = 11$; tL/B, $n = 4$; nLN, $n = 10$; mLN, $n = 7$; PE, $n = 5$, mBrain, $n = 10$. **d, e** tSNE plot of DCs, color-coded by clusters, cell subsets, and canonical marker gene expression (gray to red). **f** Partitioning of dendritic cell (DC) subsets on tSNE plot of myeloid cells in (**a**). **g** Cell number and relative proportion of DC subsets in each sample. **h** Tissue preference of DC subsets. $R_{O/E}$ is the relative score of observed cell numbers over expected cell numbers calculated by chi-square test. The $R_{O/E}$ values of all tissue origins are shown in different colors. Black dots represent different patients. *$p < 0.05$; **$p < 0.01$, two-sided Student's $t$ test. **i** Median expression of selected marker genes for DC subsets associated with their functionality in each DC subset. **, one-way ANOVA test $p$-value < 0.01. pDCs, $n = 172$ cells; Activated DCs, $n = 456$; CD1c+ DCs, $n = 1,782$; CD141+ DCs, $n = 303$; CD207+CD1a+ LCs, $n = 177$; CD163+CD14+ DCs, $n = 1,197$. **j** Representative flow cytometry plots showing pDC (CD11c-CD123+ DCs) populations in primary tumor (T09) and normal lung (N09) tissues. **k** Paired dot plot of the percentage of pDC (CD11c-CD123+ DCs) population in myeloid cells (CD45+Lin-HLA-DR+) derived from nLung-tLung paired samples (four pairs; P0009, P0014, P0019, P0041). The increase of pDC populations was detected in the selected primary tumor tissues (T09, T14, and T41). $P$-value = 0.26, two-sided Student's $t$ test. In the box plot in (**h**) and (**i**), each box represents the interquartile range (IQR, the range between the 25th and 75th percentile) with the mid-point of the data, whiskers indicate the upper and lower value within 1.5 times the IQR.

LUAD patients. Each B cell subtype displayed a slightly different B cell receptor or Ig light chain variable gene expression profile (Supplementary Fig. 7e), suggesting the generation and clonal expansion of tumor antigen-specific B cells.

T lymphocytes are the central players mediating anti-tumor immunity and are the targets of immune-checkpoint therapies. For the sub-clustering analysis of T lymphocytes, we initially collected 91,227 cells from T and NK cell clusters (Fig. 1b) sharing common transcriptome characteristics, and confidently defined 64,403 T/NK cells with a secondary cell filtration using marker gene expression (average of log-normalized expression > 2). The T/NK cell sub-clusters reflected heterogeneous cell lineages and functional states that were identified as CD8+ T (naïve, effector, exhausted), naïve CD4+ T, exhausted T follicular helper, T helper, regulatory T, and NK cells (Fig. 5c, Supplementary Fig. 8a). In accordance with previous findings[9–11], we found the depletion of NK cells and the emergence of regulatory T cells (Tregs) in the primary tumor tissues compared to normal tissues (Fig. 5d, Supplementary Fig. 8b, c). Treg cells persisted in tL/B, mLN, and mBrain, delivering a suppressive mechanism of anti-tumor immunity during tumor progression and metastasis. CD8+ T cells demonstrated a dynamic functional spectrum as that in naïve, cytotoxic, or exhausted states from the transcriptional trajectory (Fig. 5e, f, Supplementary Data 7). Exhausted CD8+ T cells were mainly collected from tumor tissues (tLung, tL/B, mLN, and mBrain), whereas cytotoxic effector CD8+ T cells were collected from nLung (Fig. 5g). Naive CD8+ T cells were mostly derived from nLN and PE. Differences in the T/NK cell subset dynamics (NK, Treg, and cytotoxic or exhausted CD8+ T cells) between primary tumor and normal lung tissues were further supported by conventional flow cytometry analysis (Fig. 5h, Supplementary Fig. 10). Altogether, the changes in cellular composition and gene expression phenotype of T cells confirmed the direction of tumor immunity towards immune suppression in LUAD.

**Inference of inter-cellular and molecular interactions**. Cellular dynamics during LUAD progression was further confirmed through chi-square tests for different tissue distributions of 40 immune and stromal cell subsets (Fig. 6a). Tumor-specific populations, such as mo-Macs, pDCs, Tregs, myofibroblasts, and tumor ECs were spread out in primary tumors and distant metastases, whereas origin-specific immune and stromal cell subsets (alveolar mac, pleural mac, microglia/mac, and FB-like cells) were specifically associated with their corresponding tissue sites. The proportions of exhausted CD8+ T cells and mo-Macs were markedly increased during LUAD progression and

metastases (Fig. 6b). In addition, increase in these two subpopulations in the tumor microenvironment was associated with the high tumor mutation burden (TMB) (Fig. 6c). As high TMB is the principal predictor of successful immune checkpoint therapy, our results support the role of mo-Macs and exhausted CD8+ T cells in the successful application of immune checkpoint therapies in advanced LUAD.

To delineate the molecular associations underlying intercellular relationships, we first constructed a cellular communication network using potential receptor-ligand pair interactions. In the tLung, we substantiated the dominant crosstalk between the tS2, a novel cancer cell state associated with LUAD progression and metastases, with myeloid or stromal cell types (Fig. 6d, Supplementary Fig. 9, Supplementary Data 8). In the network, interactions between the tS2 cells and mo-Macs were predicted to be most significant, whereas interactions between mo-Macs and exhausted CD8+ T cells were observed to be the most prominent within the immune cell network. The proportion of mo-Macs and exhausted CD8+ T cells also demonstrated positive correlations with an increase in tS2 cancer cells (Fig. 6e). For other cell types, we found potential interactions between tS2/Malignant cells and tumor ECs through angiogenesis signaling molecules, such as VEGF-VEGFRs and ephrin-Eph receptors[54] (Fig. 7a, b). Tumor ECs would receive angiogenic stimulatory signals from mo-Mac/ malignant cells through *VEGF* and its receptor *FLT1/VEGFR1*, *KDR/VEGFR2*, as a key mediator of angiogenesis in cancer[55–57] for samples of all tumor stages or for brain metastasis samples.

Further, we predicted the molecular interactions between mo-Macs, exhausted CD8+ T, and cancer cells in primary tumors (tLung and tL/B) or distant metastases (mLN and mBrain), which had a dominant crosstalk in LUAD (Fig. 7c). Most ligand-receptor pairs between mo-Macs and tS2/Malignant cells were involved in signaling of growth factors, such as *VEGFA* and *VEGFB* for all stage samples. Malignant cells would receive activation signals from mo-Macs through TNFR (*TNF-TNFRSF1A*), TGFBR (*TGFB1-TGFBR2*), and EGFR (*EREG-EGFR*) in metastatic lymph nodes. The TNFR signaling was also prominent in the late-stage tL/B. In return, malignant cells from the metastatic lymph nodes would provide the growth signal to mo-Macs (*CSF1-CSF1R*). Potential signal transduction to the exhausted CD8+ T cells was mostly inhibitory, delivered by tS2/ malignant cells for samples of all tumor stages (*NECTIN2-TIGIT*) or for metastatic lymph nodes (*LGALS9-HAVCR2*). Interestingly, mo-Macs were predicted to deliver both activating (*TNF-TNFRSF1B/ICOS*) and inhibitory (*LGALS9-HAVCR2*) signals to exhausted CD8+ T cells. Therefore, our results have demonstrated the complex nature of mo-Macs in LUAD, which greatly

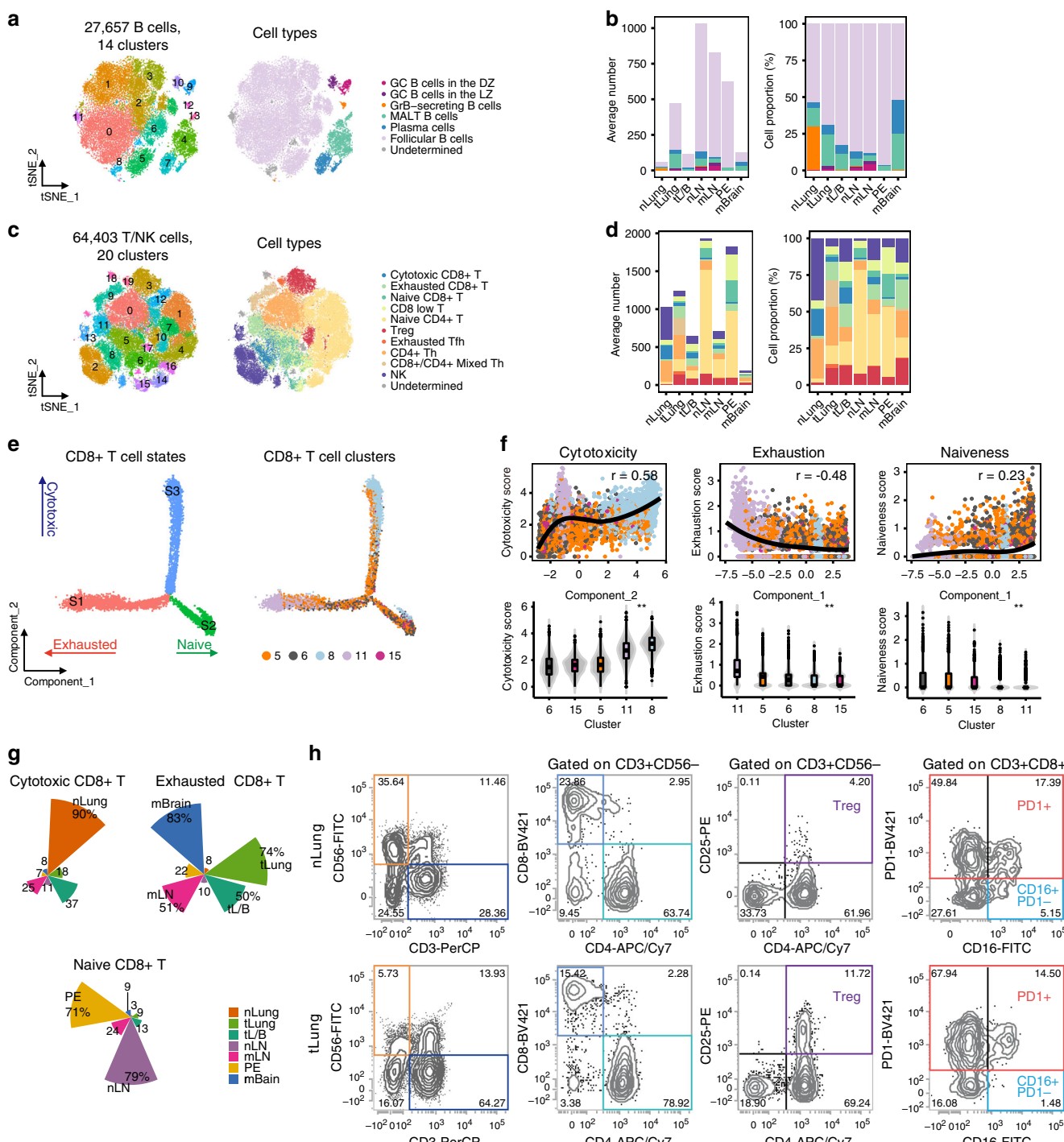

**Fig. 5 B cell- and T/NK cell-mediated immune responses during lung cancer progression. a** tSNE plot of B cells, color-coded by clusters and cell subsets as indicated. DZ: dark zone; LZ: light zone; GrB, granzyme B; MALT: mucosa-associated lymphoid tissue. **b** Average cell number and relative proportion of B cell subsets from tissues of each origin (excluding undetermined cells). nLung, *n* = 11 samples; tLung, *n* = 11 samples; tL/B, *n* = 4 samples; nLN, 10 samples; mLN, *n* = 7 samples; PE, *n* = 5 samples, mBrain, *n* = 10 samples. **c** tSNE plot of T/NK cells, color-coded by clusters and cell subsets as indicated. Tfh: T follicular helper; Th: T helper. **d** Average cell number and relative proportion of T/NK cell subsets from tissues of each origin (excluding undetermined cells). nLung, *n* = 11 samples; tLung, *n* = 11 samples; tL/B, *n* = 4 samples; nLN, 10 samples; mLN, *n* = 7 samples; PE, *n* = 5 samples, mBrain, *n* = 10 samples. **e** Unsupervised trajectory of CD8+ T cell functional state transitions. **f** Correlation of Monocle components with T cell functional features (mean expression of signature genes in Supplementary Fig. 8a). Each dot indicates single cells colored by their clusters. Solid black line and the top-right text (r) denote LOESS fit and Pearson's correlation, respectively (top). Violin plot of T cell functional features in each cluster (bottom). **, one-way ANOVA test p-value < 0.01. **g** Tissue distribution along functional states in CD8+ T cells. **h** Flow cytometry plots gated on NK and T cell subsets (Treg, cytotoxic and exhausted CD8+ T cells) from primary tumor (T08) and normal lung (N08) tissues.

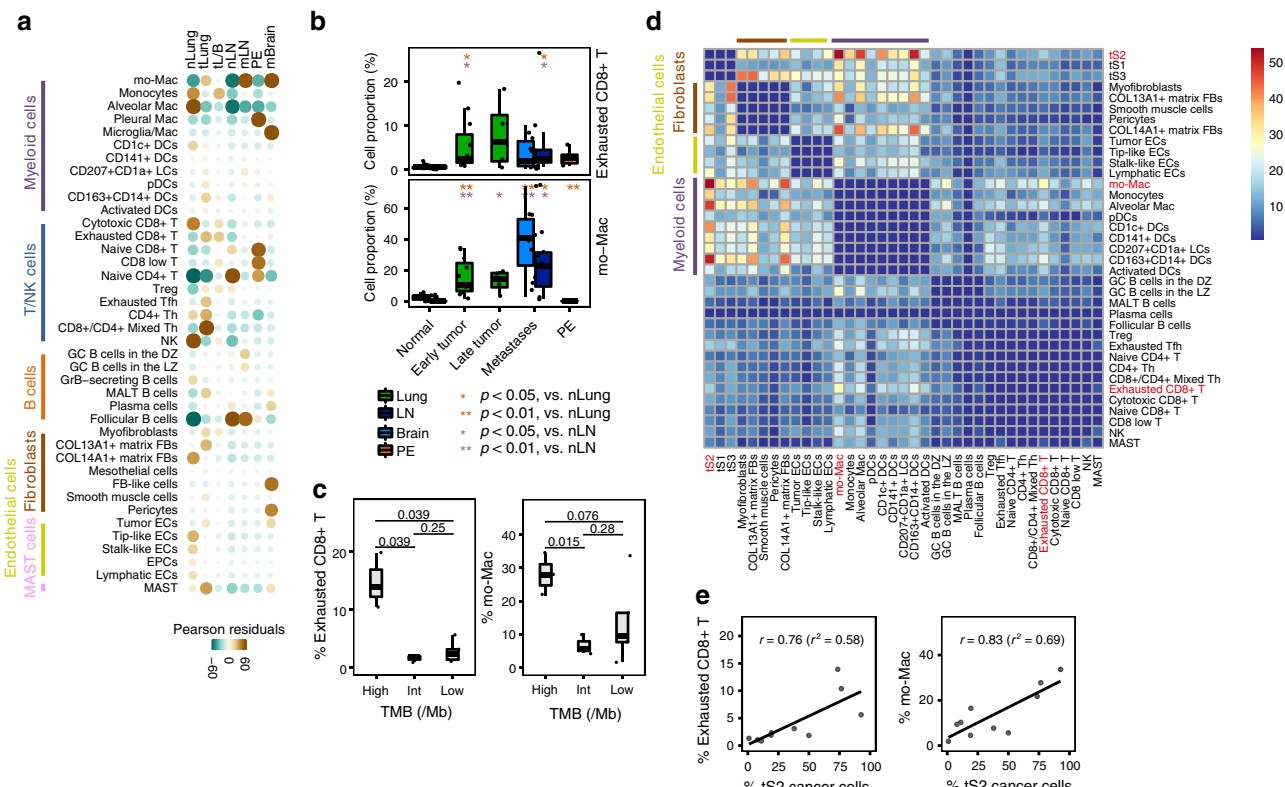

**Fig. 6 Phenotypic changes during LUAD progression and metastases. a** Tissue distribution map for each of the 40 immune and stromal cell subsets. Pearson residual calculated using the chi-square test was used to adjust cell-sampling biases between tissue origins. Brown and green colors indicate enrichment and depletion, respectively. Circle size is proportional to the contribution of a given cell. **b** Increased proportion of exhausted CD8+ T cells/ mo-Macs during LUAD progression and metastases. Immune cell proportion was estimated within a non-epithelial compartment. *$p < 0.05$; **$p < 0.01$, two-sided Student's $t$ test. nLung, $n = 11$ samples; tLung, $n = 11$ samples; tL/B, $n = 4$ samples; nLN, 10 samples; mLN, $n = 7$ samples; PE, $n = 5$ samples, mBrain, $n = 10$ samples. **c** Enrichment of exhausted CD8+ T cells/mo-Macs in tLung samples with a high mutational burden (TMB). The significance was determined using two-sided Student's $t$ test. high, $n = 3$ samples, int, $n = 3$ samples, low, $n = 5$ samples. **d** Heat map depicting the number of significant interactions between tLung cell subsets. **e** Association between proportional changes in exhausted CD8+ T cells/mo-Macs and tS2 cancer cells in tLung. The proportion of tS2 cells was estimated with respect to all malignant cells in each sample. Top-right text ($r$ and $r^2$) represents Pearson's correlation and its coefficient of determination. In the box plot in b and c, each box represents the interquartile range (IQR, the range between the 25th and 75th percentile) with the mid-point of the data, whiskers indicate the upper and lower value within 1.5 times the IQR.

influences T cell functionalities to balance immune activation and exhaustion. Taken together, the inter-cellular interactions suggest a tight relationship between immune cell dynamics and molecular features of cancer cells that may determine the prognostic and therapeutic responses in LUAD.

## Discussion

In the present study, we have depicted the cellular landscape of LUAD from the early to the advanced stages, encompassing the primary and metastatic sites. This LUAD atlas has revealed the characteristics of tumor cells and associated microenvironments, and further illuminated changes in cellular and molecular networks during tumor progression. To the best of our knowledge, this provides the most comprehensive cellular interaction map of LUAD and a framework for future discoveries of molecular and cellular therapeutic targets.

Through systemic multi-patient analyses, we have uncovered malignant molecular features of cancer cells previously masked by inter- and intra-patient genomic heterogeneity. In particular, the projection of cancer and normal epithelial cells into a joint transcriptional trajectory has revealed their similarities and disparities, implicating the paths of malignant transformation. Two transcriptional branches reflected the differentiation programs for ciliated (S3) or alveolar (S1) cells in the lung. Club cells were located at the

root of all three branches, suggesting versatile progenitor properties. Normal samples demonstrated a variable branch distribution indicating regional heterogeneity with differential proximal (ciliated and club) or distal (alveolar type) cells[58] (Supplementary Fig. 3c). On the contrary, cancer cells were found mostly in the S1 and S2 branches, showing de-regulation or complete deviation from normal epithelial transcriptional programs. The cancer cell signature at the S2 branch, tS2, was specifically associated with lung cancer progression and metastases in LUAD patients.

Intriguingly, club cells in the S1 and S3 branches expressed a touch of distal alveolar cell marker and microtubule assembly genes, respectively. This suggested the commitment towards alveolar or ciliated cells (Supplementary Data 3). On the contrary, club cells at the S2 branching point highly expressed genes involved in innate immune response and detoxification, representing their original protective function. This club cell subset might include tumorigenic progenitors susceptible to oncogenic transformation, as evidenced by sharing of S2 branch formation with tS2 cells.

We have also found that most alterations in the tumor microenvironment from normal lung tissues were inflicted at an early stage, and then sustained in later stages. First, tumor ECs acquired highly angiogenic, yet immune-compromised properties. Second, myofibroblasts gradually replaced matrix fibroblasts in the tumor stroma. Third, mo-Macs and dendritic cells (CD163

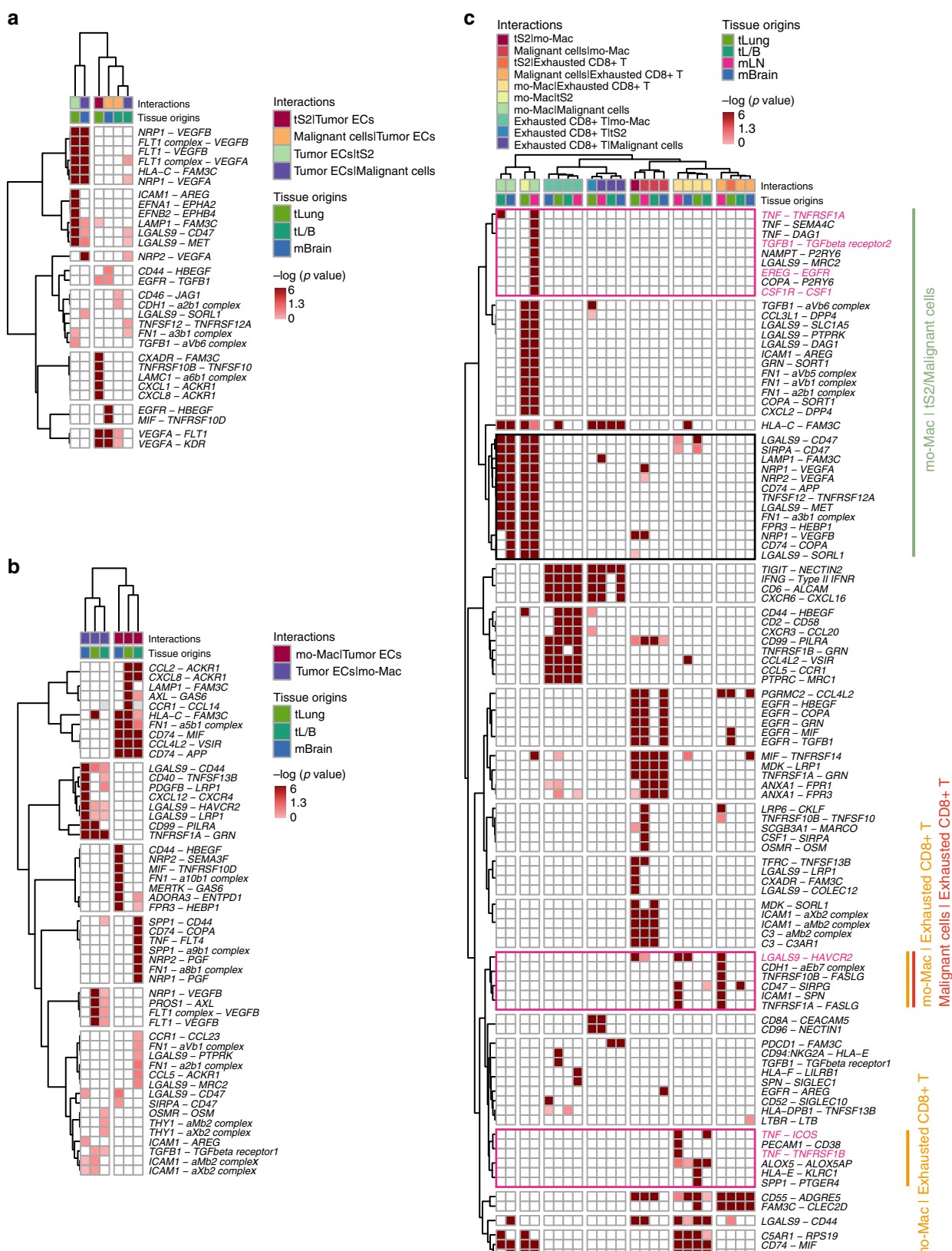

**Fig. 7 Significant ligand-receptor pair genes accounting for specific inter-cellular interactions.** Heatmap depicting significant interactions between (**a**) Tumor ECs and cancer cells; (**b**) mo-Macs and tumor ECs; (**c**) Exhausted CD8+ T cells, mo-Macs, and cancer cells in our LUAD collections (tS2 in tLung and malignant cells in tL/B, mLN, and mBrain). One-sided p-value is calculated from permutation test.

+CD14+ DCs) expanded and differentiated towards an overall anti-inflammatory phenotype, and overpowered alveolar macrophages in lung tissues and conventional DCs in the LNs. Fourth, B cells were activated and expanded in tumor tissues, suggesting humoral immune responses against tumor antigens. Fifth, cytotoxic NK cells were diminished; however, regulatory T cells were observed to be increased. Within the CD8+ T cell subsets, an exhausted T cell phenotype has expanded throughout cytotoxic effector populations. These alterations in stromal and immune populations cooperatively transformed immune-competent tissues into an immune-suppressive tumor microenvironment. Aberrant anti-tumor immune responses involving antibodies or regulatory and exhausted T cells also provide therapeutic opportunities to direct the immune reaction into productive directions using immune checkpoint inhibitors and other immune modulators.

Finally, we demonstrated vibrant cell-population dynamics and molecular interactions between the tumor, stromal, and immune compartments. Numerous highly significant interactions were inferred between mo-Macs and aggressive/metastatic tumor cells involving the activation of TNF, TGF-β, and EGFR signaling pathways in tumor cells; these interactions may induce ERK-MAPK signaling, invasive mesenchymal phenotypes, and tumor growth. Malignant cells differentially induced the activation of cellular and humoral immune responses in individual patients; however, they concomitantly provided inhibitory signals to induce immune exhaustion. This dual function of immune activation and regulation was also predicted for mo-Mac populations, as they provided both stimulatory and inhibitory signals to T cells. In conclusion, these results illustrate a complex biological picture, but also suggest a therapeutic opportunity to target mo-Macs in addition to the other T cell populations for the redirection towards immune activation.

## Methods

**Human specimens**. The present study has been reviewed and approved by the Institutional Review Board (IRB) of the Samsung Medical Center (IRB no. 2010-04-039-052), and all subjects have provided their written informed consent. Forty-four patients diagnosed with pathological LUAD were enrolled in the present study (Supplementary Data 1). The average age was 62.2 years old and 38.6% of them were female. A total of 58 samples were collected and immediately transferred for single-cell isolation. Tumor tissue, distant normal lung, normal lymph node, and metastatic brain tissues were obtained during conserving surgery at the Samsung Medical Center (Seoul, Korea) from LUAD patients that had not received prior treatment. Normal lung tissues were separated from the malignant region by at least 5 cm. Metastatic lymph node and lung tumor tissues were collected from advanced-stage LUAD patients through endobronchial ultrasound and bronchoscopy. Pleural fluids were obtained from LUAD patients through malignant pleural effusion.

A total of six tissue samples (tumor-normal pair) from three LUAD patients were additionally collected and immediately dissociated for the flow cytometry analysis. The collected tissues were as follows: LUNG_T14 (stage IIIA), LUNG_N14, LUNG_T41 (stage IIIA), LUNG_N41, LUNG_T42 (stage IA), LUNG_N42, LUNG_T43 (stage IB), LUNG_N43.

**Sample preparation**. Single-cell suspensions of the collected tissues were prepared through mechanical dissociation and enzymatic digestion within 16 h after surgery. Single-cell isolation was performed differently depending on the sample conditions. (1) Tumor and distant normal lung tissue dissociation was performed using a tumor dissociation kit (Miltenyi Biotech, Germany) following the manufacturer's instructions. Briefly, tissues were cut into pieces that were 2–4 mm in size and transferred to a C tube containing the enzyme mix (enzymes H, R, and A in RPMI1640 medium). The GentleMACS programs h_tumor_01, h_tumor_02, and h_tumor_02 were run with two 30-min incubations on the MACSmix tube rotator at 37 °C. (2) Normal lymph node tissue and biopsy samples of metastatic lymph nodes and lung tumor tissues were dissociated using collagenase/hyaluronidase (STEMCELL Technologies, Vancouver, Canada) and DNase I, RNase-Free (lyophilized) (QIAGEN, Hilden, Germany). The tissues were chopped into pieces that were 2–4 mm in size using a sterile pair of scissors, placed in a 35⁻mm dish, and incubated in an enzyme solution (collagenase/hyaluronidase (STEMCELL Technologies, Vancouver, Canada) and DNase I, RNase-Free (lyophilized) (QIAGEN, Hilden, Germany) prepared in RPMI1640 medium at 37 °C for 1 h. The tissue

pieces were re-mixed by gentle pipetting at 20-min intervals during incubation. (3) Metastatic brain tissue was chopped into pieces that were 2–4 mm in size using a sterile pair of scissors, placed in a 100-mm dish, and incubated in an enzyme solution (collagenase (Gibco, Waltham, MA, USA), DNase I (Roche, Basel, Switzerland), and Dispase I (Gibco, Waltham, MA, USA); prepared in DMEM) at 37 °C for 1 h. The tissue pieces were re-mixed by gentle pipetting at 15-min intervals during incubation. (4) Pleural fluids were transferred to a 50-ml tube, and the cells were centrifuged at 300g.

Each cell suspension was transferred to a new 50-ml (15-ml tube for biopsy samples) tube after being passed through a 70-µm strainer. The volume in the tube was readjusted to 50 ml (or 15 ml) with RPMI1640 medium, and the contents were centrifuged to remove the enzymes. The supernatant was aspirated, the cell pellet was resuspended in 4 ml of RPMI1640 medium, and the dead cells were removed using Ficoll-Paque PLUS (GE Healthcare, Chicago, IL, USA) separation.

**Single-cell RNA sequencing and read processing**. Each cell suspension was subjected to 3′ single-cell RNA sequencing using Single Cell A Chip Kit, Single Cell 3′ Library and Gel Bead Kit V2, and i7 Multiplex Kit (10x Genomics, Pleasanton, CA, USA) with a cell recovery target of 5000, following the manufacturer's instructions. Libraries were sequenced on an Illumina HiSeq2500, and mapped to the GRCh38 human reference genome using the Cell Ranger toolkit (version 2.1.0).

**Whole-exome sequencing and data processing**. Exomes of 11 formalin-fixed and paraffin-embedded lung tumor tissues and paired blood samples were captured using the SureSelectXT Human All Exon V5 kit (5190-6208, Agilent, Santa Clara, CA, USA). Sequencing libraries were constructed for the HiSeq2500 system (Illumina, San Diego, CA, USA) and sequenced using the 100-bp paired-end mode of the Hiseq PE Cluster kit v4 (PE-401-4001, Illumina, San Diego, CA, USA), and the Hiseq SBS kit v4 (PE-401-4003, Illumina, San Diego, CA, USA). Exome-sequencing reads were aligned to the hg38 reference genome using BWA-0.7.17. Putative duplications were marked by Picard (version picard-tools-2.18.22-SNAPSHOT). Sites potentially harboring small insertions or deletions were realigned, and recalibrated by employing GATK (v4.0.5.1) modules with known variant sites identified from phase 3 of the 1000 Genomes Project and dbSNP-151. GATK4 Mutect2 was used to call somatic mutations. The whole-exome sequencing (WES) coverage used was 100× for the tumors and 50× for the paired blood samples.

**Immunohistochemistry**. Patient tissue samples subjected to 3′ single-cell RNA sequencing and obtained from BioBank were fixed in 10% formalin, and embedded in paraffin. Thereafter, 4-µm-thick sections were prepared. The following antibodies and dilutions were used to detect the respective proteins: anti-IGFBP3 (mouse, 1:100, NBP2-12364, Novus Biologicals, Centennial, CO, USA), anti-CK19 (rabbit, 1:500, NB100-687, Novus Biologicals), anti-AG2 (rabbit, 1:200, NBP2-27393, Novus Biologicals), anti-S100a2 (rabbit, 1:300, ab109494, Abcam, Cambridge, UK), and anti-αSMA (Mouse, M0851, DAKO Agilent, Santa Clara, CA, USA).

**Flow cytometry**. Dissociated cells were multi-stained with three to five antibodies at 4 °C for 1 h, and then washed once with phosphate buffered saline. All antibodies were used at concentrations recommended by the manufacturer. After filtering through a round-bottom tube with a 40-µm strainer-cap, the cells were analyzed using FACSVerse and FACSuite v1.2 software (BD Biosciences, San Jose, CA, USA).

In order to examine myofibroblasts, we first stained cells with anti-human EpCAM-PerCP/Cy5.5 (Cat# 347199, BD) and CD45-PE (555483, BD) antibodies. Then cells were fixed in 2% paraformaldehyde/PBS, permeabilized in Intracellular staining Perm Wash Buffer (BioLegend, San Diego, CA, USA), and stained with the anti-aSMA-Alexa 488 (ab197240, Abcam) antibody.

The pDC populations in lung tissues were identified using anti-CD45-BV421 (368521, BioLegend) and anti-Human 4-Color Dendritic Value Bundle (340565, BD). Due to a low proportion of pDCs in the lung tissues, only samples in which CD45+Lin-HLA-DR+ cells exceed 1% of the total cell number were used for the analysis.

In order to examine NK, T, and regulatory T cells, we stained cells with FITC-conjugated anti-human CD56 (CD56-FITC, Cat# 318303), CD3-PerCP (300428), CD4-APC/Cy7 (317417), CD8-BV421 (344747), and CD25-PE (302605) antibodies. CD3-PerCP (300428), CD8a-PE (300908), CD16-FITC (360715), and PD-1-BV421 (329920) antibodies were used together to label cytotoxic or exhausted T cells.

**Filtering and normalization of scRNA-seq data**. We applied three quality measures on raw gene-cell-barcode matrix for each cell: mitochondrial genes (≤20%), unique molecular identifiers (UMIs), and gene count (ranging from 100 to 150,000 and 200 to 10,000). The UMI count for the genes in each cell was log-normalized to TPM-like values, and then used in the log2 scale transcripts per million (TPM) plus 1 (ref. [59]). For each batch, we used the filtered cells to remove genes that are expressed at low levels by counting the number of cells (min.cells) having expression of each gene $i$, and excluded genes with min.cells < 0.1% cells. For the remaining cells and genes, we defined relative expression by centering using

the Seurat ScaleData function with the following parameters: do.scale = FALSE, do.center = TRUE, scale.max = 10. The relative expression levels across the remaining subset of cells and genes were used for subsequent analysis.

**Unsupervised dimensional reduction and clustering**. Variably expressed genes with mean expression between 0.0125 and 3 and quantile-normalized variance greater than 0.5 were selected using Seurat v2.3.4 (ref. [60]) (https://satijalab.org/seurat/) in R version 3.4, and then used to compute the principal components (PCs). A subset of significant PCs was selected using the JackStraw and PCElbowPlot functions of Seurat. Cell clustering and tSNE visualization were performed using the FindClusters and RunTSNE functions, respectively. The annotations of cell identity on each cluster were defined by the expression of known marker genes.

We compared the results for cell lineage annotations based on independent cell clusters from different clustering methods as follows: Seurat, CIDR[61], RCA[62], and SC3 (ref. [63]) (Supplementary Fig. 1b). The clustering by Seurat algorithm showed high agreement with the results obtained using the other methods.

**Inference of CNV from scRNA-seq data**. In order to separate malignant tumor cells from non-malignant cells, CNV aberrations were inferred from the perturbation of chromosomal gene expression. First, we adjusted the proportion of putative malignant cells below 20%, by adding normal cells derived from normal lung tissues to the individual tumor samples collected from tLung, tL/B, mLN, and mBrain. Secondly, we filtered out less informative genes that were expressed in less than 10 cells and that had a mean expression of less than 0.1 across all cells at the $\log_2$ scale. Third, the expression of each gene was transformed into a Z-score on a limited scale from $-3$ to 3. Fourth, after sorting the genes by their chromosomal position, the signal for CNV was estimated using the moving averages of 100 genes as the window size within individual chromosomes, and adjusted using centered values across genes[64]. Finally, we summarized the CNV signal with two parameters, including mean squares of estimates across all windows in the $x$-axis, and correlation of the CNV of each cell with the average of the top 5% of cells in the $y$-axis[8]. Cancer cells showing perturbation in their CNV signal (>0.02 mean squares or >0.2 CNV correlation) were classified as malignant. The codes for CNV inference are available at the Github repository (https://github.com/SGI-LungCancer/SingleCell).

**Prediction of tumor mutation burden**. The tumor mutation burden (TMB) was estimated from bulk WES data for 11 formalin-fixed and paraffin-embedded lung tumor tissues. The TMB is defined as the total number of mutations per coding area of a tumor genome. The TMB status of tLung samples was determined by a range of TMB values: high TMB, >30 mutations/Mb; int, intermediate; low TMB, <25 mutations/Mb.

**Marker gene selection specific to cancer cells**. For pairwise comparisons between early- versus advanced-stage primary, or primary versus metastatic cancer cells, we filtered out non-malignant cells among cancer cells from tLung, tL/B, mLN, and mBrain. A total of 6352, 6400, 2961, and 15423 malignant cells, respectively, were used to identify the differentially expressed genes. Malignant cells from tLung were used as a control group to evaluate the advanced stage- and metastasis-specific gene regulation. We manually calculated the $\log_2$ fold change ($\log_2$FC) between two groups (tL/B, mLN, mBrain versus tLung) using the Seurat FindMarkers function. The significance of the difference was determined using two-sided Student's $t$ test with Bonferroni correction. Signature genes were required to be expressed in >25% of cells within either of the two cell groups (marked as PCT, percentage of cells). Genes were selected as signatures based on the statistical threshold (absolute $\log_2$FC > 0.585, two-sided Student's $t$ test $p$-value<0.01, and adjusted $p$-value (Bonferroni) <0.01).

**Inference of tumor cell state using trajectory analysis**. We first extracted the epithelial cell clusters (Fig. 1b) from the single-cell RNA sequencing data on nLung samples. In order to generate a trajectory, we adopted only malignant cells, as estimated by CNV inference, among the epithelial cells in the tLung samples. We employed the Monocle (version 2) algorithm[14] using variable genes selected by Seurat as the input to determine the differential tumor cell states referenced against normal epithelial cells. The gene-cell matrix in the scale of UMI counts was provided as input to Monocle, and then, its newCellDataSet function was called to create an object with the parameter expressionFamily = negbinomial.size. The epithelial cell trajectory was inferred using default parameters of Monocle after dimension reduction and cell ordering.

**Inference of the developmental trajectory for immune cells**. The cell state transitions for monocytes/mo-Macs and T cells were estimated using the Monocle (version 2) algorithm[14]. For monocytes/mo-Mac, we first selected single cells in clusters defined as monocytes and monocyte-derived macrophages (mo-Macs) from the whole dataset. We then generated separate trajectories for lung tissues and lymph nodes. For CD8+ T cells, we took single cells expressing CD8A and CD8B (average of log-normalized expression > 0) within the CD8+ T cell clusters (T-C5, 6, 8, 11, and 15). Similar to the trajectory analysis for tumor cells, the gene-cell

matrix in the scale of UMI counts was provided as an input to Monocle, and then, its newCellDataSet function was called to create an object with the parameter expressionFamily = negbinomial.size. For the macrophages, variably expressed genes were selected to have normalized variance over the fitting curve and mean expression greater than 0.001, as estimated by the Monocle dispersionTable function. For CD8+ T cells, variably expressed genes selected by Seurat that had a mean expression between 0.0125 and 3 and quantile-normalized variance larger than 0.5 were used as inputs. The cell trajectory was inferred using default Monocle parameters after dimension reduction and cell ordering.

**Identification of signature genes**. We applied the Seurat FindAllMarkers function to identify specific genes for each cell subset. For the selection of marker genes specific to each cell state (as estimated by the trajectory analysis), we calculated the $\log_2$ fold change ($\log_2$FC) between two groups (a cell state/subset vs. other cells) using the Seurat FindMarkers function. The significance of the difference was determined using two-sided Student's $t$ test with Bonferroni correction. Signature genes were required to be expressed in >25% of cells within either of the two cell groups (marked as PCT, percentage of cells). Genes were selected as signatures based on the statistical threshold ($\log_2$FC > 1, two-sided Student's $t$ test $p$-value < 0.01, and adjusted $p$-value (Bonferroni) < 0.01). The selected genes were categorized according to the functional gene sets in the Gene Ontology (GO) Biological Process using database for annotation, visualization and integrated discovery (DAVID)[65,66] (https://david.ncifcrf.gov/) pathway enrichment analysis.

**Survival analysis**. RNA-seq and clinical data from patients' LUAD and squamous cell carcinoma (LUSC) samples were obtained from the Cancer Genome Atlas (TCGA) to evaluate the prognostic effects of gene sets derived from specific cell states. The RNA-seq data (Level 3) included 494 LUAD and 490 LUSC (updated in 2017) tumors, and the expression of each gene was represented using the log2 (TPM + 1) scale. We acknowledged patient survival if the time of death after diagnosis was longer than 10 years for a more refined analysis of survival rate. The tumor samples were divided into two classes along the 25th and 75th percentiles of the mean expression of the target genes. Survival curves were fitted using the Kaplan–Meier formula in the R package 'survival', and visualized using the ggsurvplot function of the R package 'survminer'.

**Gene–gene functional association network**. We constructed a network between marker genes specific to tumor ECs to represent their functional association. The network was quantified using the Jaccard index, known as an intersection over the union, between the GO Biological Process terms for all possible pairs of genes. A total of 68 genes (22 upregulated and 46 downregulated genes) were identified as marker genes significantly expressed in tumor ECs in comparison to all other ECs (absolute $\log_2$FC > 1, two-sided Student's $t$ test p-value<0.01, adjusted p-value (Bonferroni) < 0.01, and PCT > 0.25). For visual clarity, we constructed the gene–gene network using only associations with a Jaccard index exceeding 0.05, for a total of 56 genes (18 upregulated and 38 downregulated genes). This network was visualized with the force-directed layout algorithm in the open-source platform Cytoscape (version 3.5.1)[67].

**Cell–cell interaction network**. We mapped the receptor-ligand pairs using CellPhoneDB (www.cellphonedb.org)[68,69] onto our cell subsets within tissues of each origin to identify cell–cell interactions. This method infers the potential interaction strength between two cell subsets based on gene expression level, and provides the significance through permutation test (1000 times). Only receptors and ligands expressed in more than 25% of the cells in the specific cell subsets were considered in each run. The resulting adjacency matrices were generated for all cell–cell interactions within our collection. For the analysis, we applied four filtering steps to the raw adjacency matrices: (1) interaction pairs with collagens were removed, (2) cell–cell interactions within identical cellular lineages were excluded, (3) only cell subsets defined in more than 0.1% of the cells in immune and stromal cells were analyzed, and (4) and significant interactions (satisfied with one-sided $p$-value for a permutation test <0.05) were identified between cell subsets in tissues of each origin.

**Reporting summary**. Further information on research design is available in the Nature Research Reporting Summary linked to this article.

## Data availability
Processed data can be accessed from the NCBI Gene Expression Omnibus database (accession code GSE131907). Raw single-cell RNA sequencing and bulk WES data are available in the European Genome-phenome Archive database (accession code EGAD00001005054). Single-cell expression data can be explored online at http://urecasinglecell.kr.

## Code availability
The codes generated during this study are available at Github repository (https://github.com/SGI-LungCancer/SingleCell).

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

## Acknowledgements

The authors thank research fellows at the Samsung Genome Institute for their valuable comments and the Korean Bioinformation Center (KOBIC) for technical support to build and manage the data visualization website (http://ureca-singlecell.kr). We thank the Samsung Medical Center BioBank for providing the biospecimens that were used in the immunohistochemistry. The results shown here are in part based upon data generated by the TCGA Research Network: https://www.cancer.gov/tcga. This research was supported by the Collaborative Genome Program for Fostering New Post-Genome Industry of the National Research Foundation (NRF) funded by the Ministry of Science and ICT (MSIT) (NFR-2017M3C9A6044633 to M.A. and NRF-2017M3C9A6044636 to H.L.).

## Author contributions

M.A. and H.L. designed and supervised the study. N.K. performed data analysis, with significant contributions from Y.H. H.K. and K.L. coordinated sample collection and clinical data interpretation with help from B.K., J.H.C., J.W.C., J.L., and Y.S. Y.H. built the website. H.E. and S.C. performed flow cytometry analyses. Y.C. performed immunohistochemical analyses. W.P., H.J., J.S., S.L., J.A., K.P., and J.J. contributed to critical data interpretation. H.L. and N.K. wrote the manuscript, and all authors contributed to the writing and provided comments.

## Competing interests

The authors declare no competing interests.
