## [Peer Review File · Nature Communications]

Reviewers' comments:

Reviewer #1 (Remarks to the Author):

In this study, Nayoung et al. performed single-cell RNA sequencing on 208,506 cells from 44 patients with metastatic lung adenocarcinoma. Unsupervised clustering revealed distinct epithelial, endothelial and immune cell subtypes. The authors also identified a new cancer cell subtype dominating the metastatic stage and illustrated the functions of some immune cell subtypes including mo-Macs, dysfunctional pDCs, NK cells and GC B cells in tumor microenvironment. To my knowledge, this manuscript represents the most comprehensive description of cell population associated with human metastatic lung adenocarcinoma, both from the perspective of cell numbers and patient numbers. Overall, this study provides a valuable and rich resource for understanding the immune landscape in tumors, but there are notable drawbacks of this study. Very little attention was paid to the connection between primary sites and metastasis sites, thus there are glaring holes regarding the similarity or diversity of immune cells or the lineage development of malignant cells between tLung and mBrain.

Major comments

1. In line 107-109 the authors mentioned that they excluded non-malignant cells when performing trajectory analysis. It is very confusing that the different numbers of epithelial cells were labeled in Extended Data Fig.2a and 2i. Regardless, a more clear and detailed method about the trajectory analysis should be provided. The authors also need to do the trajectory analysis of epithelial cells patient-by-patient (only in patients with sufficient cells) due to the high heterogeneity of malignant cells.
2. In Fig.2d, the authors analyzed the relationships between the proportions of tS2 cells and clinical parameters. It seems that that poor malignant cells were collected in a few of patients, thus it is key to perform the same analysis using the proportions of malignant cells, and determine the differences among malignant cells in tS1, tS2 and tS3.
3. It is not clear how the authors defined tumor-derived ECs (line 160), and what the interactions are like between this cluster and malignant cells or macrophages. It's known that tumor-associated macrophages could promote blood vessels to grow and invade the tumor to feed it.
4. Many conclusions in this paper are based on cell subtype proportion changes, such as enrichment with myofibroblasts (line 175-177) or pDC (line 219-222). This process (calculating the cell proportions using scRNA-seq can be error-prone due to the low cell numbers collected in some samples. The authors need to do FACs and statistical analysis using each patient's data separately to validate some most important results of proportion changes.

5. From the heatmap in Fig.4b, the expression pattern of pleural macrophage looks very similar to monocytes. What are the specific markers of pleural macrophages besides tissue enrichment? And what are the potential functions or relationships between pleural macrophage and LUAD?
6. In Extended Data Fig.5d, the trajectory analysis separated macrophages into two branches. Differences between macrophages in S2 and S3 should be analyzed.
7. The evidences for the existence of CD14+ preDCs were weak. It is very close to the monocytes and macrophages in the tSNE plot, and the authors didn't label a specific marker of this cluster.
8. Statistical tests are very weak throughout and in some cases completely missing. Here are some examples:
 - a) Line 80: "we confirmed T and B lymphocyte enrichment and the decline of NK and myeloid cells...".
 - b) Line 127: "Enrichment in tS2 cells...".
 - c) Line 195: "lung tumor and distant metastasis tissues were strongly enriched in monocyte-derived macrophages".
 - d) Line 228: "Primary tumors were intriguingly enriched in B cells" (also a grammatical mistake).
9. In line 94, the authors claimed that (epithelial) sub-clusters were largely distinguished by tumor or normal tissue origin. However, all epithelial cells from normal tissue were actually overlapped with malignant cells in the tsne plot (Extended Data Fig. 2a).
10. The authors re-classified DCs into 6 subsets (line 216) as indicated by Villani et al. However, the six identified DC subtypes in that Science paper were collected from PBMC and some of these subtypes have been proven to be not DCs (Dutertre et al, Immunity, 2019). Perhaps this should be re-evaluated?
11. The authors identified 14 B cell clusters (line 230) but did not annotate these subtypes. Are these biologically meaningful clusters or only technical products driven by batch effects or algorithms? Additional discussion of these subtypes should be included.

Minor points:

1. This manuscript should be edited by native speakers as there are many language mistakes.
2. The conclusion in line 87 is not precise: "these cellular compositions reflected original tissue microenvironments and gross alterations inflicted by tumor growth and invasion". Single-cell RNA sequencing could only obtain a snapshot and has varied preferences for capturing different cell types, thus it cannot reflect original tissue microenvironment.
3. How the cells are annotated should be clearly documented

4. Detailed calculation of CNV correlation and relevant procedures for identifying malignant cells should be clearly described in Methods.

5. The order of some of the Extended Data Figs. described in the main text is off. For example: Lines 102 and 105: the description of Extended Data Fig. 2h appears before Extended Data Fig. 2g.; Line 138: Extended Data Fig. 3e -> Extended Data Fig. 3b

6. The description in line 107 is confusing: "We performed unsupervised trajectory analysis (Fig. 2a) after excluding non-malignant cells". Why did these non-malignant appear in Fig. 2a after their removal?

Reviewer #2 (Remarks to the Author):

In their paper, Kim et al. present a detailed study of the cellular environment of lung adenocarcinoma, using single cell RNA-seq (Chromium 10X 3'end tag sequencing). In contrast to previous single cell lung cancer profiling papers, the authors of this manuscript concentrate on the changes that occur in metastatic disease, which is of course responsible for most mortality associated with lung cancer.

They generate a large data set of around 250,000 cells that is very heavily immune enriched. The authors carry out detailed computational analysis of the different cell populations that were retrieved, namely epithelial cells, fibroblasts and endothelial cells and different immune cell lineages. For the cancerous epithelial cells the authors identify three distinct cancer states and derive gene signatures for each of these and show that tumor signature 2 is associated with reduced patient survival.

They examine endothelial and fibroblast states and identify distinct sub-populations that differ in their abundance depending on the origin of the cells. (e.g normal, versus tumour, versus LN from metastatic cancer). They further home into the abundance of specific myeloid and macrophage cell states, as well as the distribution of distinct B and T lymphocytes and NK cells.

Overall, they have reasonable representation of epithelial cells, but only relatively small contributions of endothelial and fibroblasts. As expected only the lung tissue itself contributes to fibroblasts and endothelial cells. A very large proportion of the cells they retrieve are immune cells. However, immune enrichment is seen for many dissociation protocols, so this is not unusual.

Lastly, they examine the tissue distribution of immune and stromal cells. They find that exhausted CD8+ T/mo-Mac cells are enriched in metastasis associated tissue. The authors also examine possible receptor-ligand interactions between different cell populations and find that interactions of tS2 signature malignant cells appear to be particularly high with myeloid cells and suggest particular receptor-ligand pairs that may be relevant.

This is a very comprehensive study that is of great interest for better understanding the interplay between the immune system and developing and evolving cancer cells and I believe this study will be of great interest to many readers.

However, there are a number of points the author should address to clarify their analysis or place their results more clearly into the context of the existing literature.

1. As already alluded to the cell purification method appear to enrich for immune cells. In this particular study this is of course very welcome. However, the work would benefit from also having some data that highlights what the actual cell type distribution is. This could be done by cell staining of tissue sections or by single nuclear sequencing of some remaining tumour section. If no more tumour is available the authors should at least highlight that the described cell type distribution is unlikely to reflect the “real” cell type distribution.

2. Figure 2a: The authors say: “We performed unsupervised trajectory analysis (Fig. 2a) after excluding non-malignant cells to adjust for inter-patient genomic heterogeneity and to find key cancer gene expression programs (Extended Data Fig. 2i).”

However, the trajectory that is shown in Fig 2a clearly contains malignant and non-malignant cells. The author must explain this more clearly.

3 The authors derive three tumour cell gene expression signatures according to the three branches of the pseudo time trajectory. They go on to say that gene signature S2 is associated with poor outcome in LUAD. I feel the authors need to discuss how this relates to previously defined LUAD cancer gene signatures. Does this gene signature overlap with those previously described? How well does S2 correlate with poor outcome? Would a classic proliferation signature do better?

Suppl Fig 3 is meant to explain this further. However, there seems to be no colour code for Fig S3a. What are the colours in the first tSNE? It is labelled with tS1, tS2 and tS3, but the distribution of these clusters is very different from tS1-3 as colour coded in the tSNE for cell states. The authors must clarify this.

4. The authors state: “The efficacy of targeted agents against EGFR signaling demonstrated minimal association with intra- and inter-tumoral heterogeneity” but must make clear that is predicted rather than actual efficacy.

5. Figure 3d: It is not clear exactly what this figure is depicting. There is very little explanation.

“The most significantly upregulated genes in tumor ECs were associated with angiogenesis, and strong activation of VEGFR and Notch signaling (Fig. 3b,d, Supplementary Table 4).”

Could the authors include some explanation of how their functional association network was derived.

6. There are extensive tables with gene signatures, but I think it would be helpful to have a supplementary table, but ideally figure that indicates the marker genes that were used to annotate each cell type. (apologies if I have overlooked such a figure.). This will make it much easier to other groups to relate their work to this study.

7. minor point: in Figure 5b exhausted is mis-spelt.

The computational packages used in the analysis employ a wide range of statistical tests. I am afraid I am not able to comment on the suitability and correct application of these tests.

Reviewers' comments:

Reviewer #1 (Remarks to the Author):

In this study, Nayoung et al. performed single-cell RNA sequencing on 208,506 cells from 44 patients with metastatic lung adenocarcinoma. Unsupervised clustering revealed distinct epithelial, endothelial and immune cell subtypes. The authors also identified a new cancer cell subtype dominating the metastatic stage and illustrated the functions of some immune cell subtypes including mo-Macs, dysfunctional pDCs, NK cells and GC B cells in tumor microenvironment. To my knowledge, this manuscript represents the most comprehensive description of cell population associated with human metastatic lung adenocarcinoma, both from the perspective of cell numbers and patient numbers. Overall, this study provides a valuable and rich resource for understanding the immune landscape in tumors, but there are notable drawbacks of this study. Very little attention was paid to the connection between primary sites and metastasis sites, thus there are glaring holes regarding the similarity or diversity of immune cells or the lineage development of malignant cells between tLung and mBrain.

Author's Responses: We appreciate the reviewer's acknowledgement of the importance of our study. Our dataset provides a valuable resource for the understanding of advanced metastatic lung adenocarcinoma. In the revision, we have paid more attention to describe the similarities and differences between the early and advanced metastatic stages of lung adenocarcinoma, focusing on the cancer cell signatures and other cell populations in the surrounding microenvironment. The aggressive cancer cell signature (tS2) was associated with tumor progression, including metastasis in the brain. Gross alterations in tumor microenvironment from normal lung tissues were observed at the early and metastatic stages.

We have also performed pairwise comparisons between cancer cells from the primary (tLung) and metastatic sites (local, tL/B; lymph node, mLN ; brain, mBrain) and listed the differentially expressed genes (revised Supplementary Figure 4a and Supplementary Table 4).

We will continue the comparative analyses between the tLung and mBrain in future studies, with additional sample preparations using non-tumor brain tissues for the comparison of background populations, and lung tumor tissues with concomitant brain metastasis. We believe that understanding of brain metastasis is particularly important to cope with the clinical courses of lung cancer.

Major comments

1. In line 107-109 the authors mentioned that they excluded non-malignant cells when performing trajectory analysis. It is very confusing that the different numbers of epithelial cells were labeled in Extended Data Fig.2a and 2i. Regardless, a more clear and detailed method about the trajectory analysis should be provided. The authors also need to do the trajectory analysis of epithelial cells patient-by-patient (only in patients with sufficient cells) due to the high heterogeneity of malignant cells.

Author's Response: This confusion arose from our dual use of "non-malignant cells" to describe epithelial cells isolated from normal tissues as well as CNV (copy number variation, inferred from gene expression)-negative cells isolated from tumor tissues. As illustrated in the 'Reviewer's ONLY flowchart', we have performed a step to separate "malignant" cells with inferred CNV aberrations, and "non-malignant" cells without CNV during the tumor cell analysis.

In the revision, we have simplified the analysis workflow in order to focus on the "malignant cells with patient-specific CNV patterns". Accordingly, Supplementary Figures 2e and 3a in the original submission have been removed.

Reviewer's ONLY Figure 1: Flowchart for the cancer cell analysis scheme.

Author’s Response to the request of trajectory analysis for each patient

Following the reviewer’s suggestion, we have performed trajectory analysis using data from a single patient for three patients with the highest number of epithelial tumor cells (LUNG_T34, LUNG_T18, and LUNG_T28) and presented them as revised Supplementary Fig. 3e. Some patients (P0034 and P0028) did not have tS3 cells, and thus did not form a separate tS3 branch. Nonetheless, the individual patient-by-patient trajectories recapitulated the separation of tS2 from normal epithelial or other tumor cell clusters.

Revised Supplementary Fig. 3e

2. In Fig.2d, the authors analyzed the relationships between the proportions of tS2 cells and clinical parameters. It seems that that poor malignant cells were collected in a few of patients, thus it is key to perform the same analysis using the proportions of malignant cells, and determine the differences among malignant cells in tS1, tS2 and tS3.

Author’s Response: In our article, the circle size represents proportions and the filled colors represent numbers (Fig. 2d). We have attached an alternative figure that separately presents numbers and proportions among malignant cells (Reviewer’s ONLY Fig. 2d, right; compared to the original format on the left). We have preferred the original bubble chart that effectively highlights the differences among malignant cells in tS1, tS2, and tS3.

Reviewer’s ONLY Figure 2d: Numbers and proportions of normal epithelial and malignant cells among cell states within tLung and nLung.

Left: original version

Right: alternative version

3. It is not clear how the authors defined tumor-derived ECs (line 160), and what the interactions are like between this cluster and malignant cells or macrophages. It's known that tumor-associated macrophages could promote blood vessels to grow and invade the tumor to feed it.

Author's Response: "Tumor-derived ECs" refer to the EC clusters mainly derived from the primary tumors and brain metastases. These endothelial cells have shown a differential gene expression, with an upregulation in angiogenesis and downregulation in immune response (PMID29988129) than the EC clusters enriched in normal tissues. We added a short description for the tumor-derived ECs in the revised manuscript as follows in lines 169-170:

"... one distinct cluster was identified as tumor-derived ECs (EC-C1) present in tLung and mBrain samples (Fig. 3c, Supplementary Fig. 5b, c)."

Further, to depict crosstalk between tumor ECs and malignant cells or between tumor ECs and mo-Macs, we have provided additional figures for potential ligand-receptor interactions (revised Fig. 7a, b). Most ligand-receptor pairs between tS2/Malignant cells and tumor ECs were associated with angiogenesis signaling molecules, such as VEGF-VEGFRs and ephrin-Eph receptors (PMID22866201). As expected, tumor ECs received angiogenic stimulatory signals from mo-Macs/malignant cells for VEGF and its receptor *FLT1/VEGFR1*, *KDR/VEGFR2*, as key mediators of angiogenesis in cancer (PMID16301830, PMID16336962, PMID24314323) for samples of all tumor stages or for brain metastases. We have included relevant descriptions of these aspects in the revised manuscript (Lines 322-327).

4. Many conclusions in this paper are based on cell subtype proportion changes, such as enrichment with myofibroblasts (line 175-177) or pDC (line 219-222). This process (calculating the cell proportions using scRNA-seq can be error-prone due to the low cell numbers collected in some samples. The authors need to do FACs and statistical analysis using each patient's data separately to validate some most important results of proportion changes.

Author's Response: We fully agree with the reviewer that scRNA-seq data can be error prone in the estimation of cellular proportions as they underestimate epithelial and stromal cells, but overestimate immune cells [nicely presented in the Supplementary figure 2 of PMID29988129]. Due to these biases, the scRNA-seq data need to be interpreted in caution and validated by independent methods.

In the present study, we have used cluster dynamics to depict phenotypic changes within a cell type. For example, myofibroblast cluster was enriched within the fibroblast cell type, and pDC cluster within the dendritic cell type for tumor tissues, compared to the normal tissues. These cluster dynamics within a particular cell type may not be as error-prone as the proportions of global cell types, such as epithelial, stromal, and immune cells. Nonetheless, the cluster dynamics also need to be validated in more patients and using other methods to draw solid conclusions.

In the revision, we have provided two additional pieces of data for the validation of cluster dynamics. Firstly, the expression of alpha-SMA (*ACTA2* gene), a representative marker for myofibroblasts, was analyzed through immunohistochemistry to demonstrate tissue localization, and through flowcytometry for the statistical analysis (revised Fig. 3h-j). The second is the flow cytometry data for detecting pDC populations as $\text{Lin}^- \text{CD11c}^- \text{HLA-DR}^+ \text{CD123}^+$ (revised Fig. 4j and k). These data confirmed the increase in myofibroblasts and pDCs, respectively, in tumor tissues compared to the case for the normal tissues.

Revised Fig. 3h-j

Revised Fig. 4j,k

5. From the heatmap in Fig.4b, the expression pattern of pleural macrophage looks very similar to monocytes. What are the specific markers of pleural macrophages besides tissue enrichment? And what are the potential functions or relationships between pleural macrophage and LUAD?

Author's Response: In Fig. 4b, "pleural macrophages" refer to those isolated from the malignant pleural effusion. As individual macrophage clusters (clusters #8 and #9) were formed, which were separate from the monocyte cluster (cluster#3), we labeled them as a distinct population. As the reviewer has pointed out, "pleural macrophages" share gene expression patterns with monocytes with regard to high levels of *FCN*, *S100A8*, and *S100A9*. However, they differ from the monocyte cluster due to the absence of the expression of pro-inflammatory cytokine genes such as *IL1B* and *CXCL8*, and the presence of *CD163* transcripts associated with the non-inflammatory M2 phenotype. They also express higher levels of *FCGR1A* (CD64) high-affinity type I IgG receptor gene involved in the Fc receptor-mediated phagocytosis compared to the monocyte cluster. In cancers, including LUAD, non-inflammatory macrophage phenotype is associated with tumor progression.

Due to the limited space in the article, we have constructed a web-based exploration site at http://ureca-singlecell.kr/go_search to provide extensive gene expression data regarding different clusters. The website can be accessed with a temporary password (nayoone.kim@sbi.co.kr) and will be freely available upon publication.

6. In Extended Data Fig.5d, the trajectory analysis separated macrophages into two branches. Differences between macrophages in S2 and S3 should be analyzed.

Author's Response: In the revised Supplementary Fig. 6e, f, we have replaced the trajectory figures with a gene expression map for the S1, S2, and S3 states in the lungs and lymph nodes. The revised figure will provide more information regarding the differences between the trajectory branches.

7. The evidences for the existence of CD14+ preDCs were weak. It is very close to the monocytes and macrophages in the tSNE plot, and the authors didn't label a specific marker of this cluster.

Author's Response: We have used a clustering-based approach for the cell type identification. In Fig. 4a,b depicting myeloid cell sub-clusters, cluster #3 was observed to express the highest levels of the monocyte marker genes *FCN*, *S100A8*, and *S100A9*. Comparatively, clusters #4, #14, and #12 were observed to express various combinations of the dendritic cell (DC) marker genes *CLEC10A*, *CD1C*, *CLEC4C*, *PTCRA*, *CCR7*, and *LAMP3*. Based on the clustering results and marker gene expression, we have labeled the cluster #3 as monocytes and clusters #4/#14/#12 as DCs.

To delineate the diversity of DC populations, we have performed further subclustering analysis using only the presumed DC clusters #4/#14/#12. As shown in Fig. 4e, most DC subclusters have shown concordant gene expression characteristics previously assigned to DC subsets [PMID28428369 and 24744755].

An exception is the cluster we termed CD14+ preDCs in the original submission, which failed to show clear DC lineage markers, and was thus named as preDCs. As per the reviewer's suggestion, to re-define their identity, we have compared DC cluster genes from Villani's [PMID28428369] and Dutertre's [PMID31474513] study, and the present study. In the results, we have found CD163+CD14+ cDC2s (Dutertre's annotation) similar to our "CD14+ preDCs", which showed the expression of the referred marker genes (cDC2: *LMNA*, *CDKN1A*, *F13A1*, *FCER1A*; CD163+CD14+ cDC2: *S100A8*, *S100A9*, *CD14*). Accordingly, we have changed the annotation to CD163+CD14+ DCs.

8. Statistical tests are very weak throughout and in some cases completely missing. Here are some examples:

- a) Line 80: "we confirmed T and B lymphocyte enrichment and the decline of NK and myeloid cells..."
- b) Line 127: "Enrichment in tS2 cells..."
- c) Line 195: "lung tumor and distant metastasis tissues were strongly enriched in monocyte-derived macrophages".
- d) Line 228: "Primary tumors were intriguingly enriched in B cells" (also a grammatical mistake).

Author's Response: We have included statistical test results in an appropriate format throughout the revised manuscript (two-sided Student's t-test, Fig. 3j, Fig. 4h and k, Fig. 6b and c, Supplementary Fig. 2d, Supplementary Fig. 5c and e, Supplementary Fig. 6b, Supplementary Fig. 7d, Supplementary Fig. 8c; two-sided Wilcoxon test, Fig. 2e; one-way ANOVA test, Fig. 2f, Fig. 4i, Fig. 5f).

To address significant changes in cell proportions, such as a) and c), we have added the statistical analyses (two-sided Student's t-test) based on the $R_{O/E}$ values. $R_{O/E}$ represents the ratio of observed cell numbers to the random expectation calculated using chi-square test [PMID29942094]. (revised Fig. 4h, Supplementary Fig. 2d, Supplementary Fig. 5c and e, Supplementary Fig. 6b, Supplementary Fig. 7d, Supplementary Fig. 8c)

a) **Revised Supplementary Fig. 2d**

b) We have removed the sentence “Enrichment in tS2 cells was partially associated with smoking status” from the original manuscript.

c) Revised Supplementary Fig. 6b

d) We have revised the sentence “Primary tumors were intriguingly enriched in B cells” as “Relative proportion of B cells was observed to be increased in primary tumors, compared to the nLung samples (Supplementary Fig. 1c, d)” in the revised manuscript (Lines 259-260).

9. In line 94, the authors claimed that (epithelial) sub-clusters were largely distinguished by tumor or normal tissue origin. However, all epithelial cells from normal tissue were actually overlapped with malignant cells in the tsne plot (Extended Data Fig. 2a).

Author’s Response: Epithelial cells dissociated from tumor tissues contain both malignant and non-malignant populations. The non-malignant population forms multi-patient clusters together with the normal epithelial cells isolated from normal tissues. The reviewer has pointed out this feature in the Supplementary Fig. 2a of the original submission.

To briefly go over the cancer cell analysis section, we initially extracted 7,270 epithelial cells from primary tumors and 3,703 cells from distant normal lung tissues. As pointed out by the reviewer, we have found some epithelial cells from primary tumors that overlapped with normal epithelial cells in the tSNE plot (original Supplementary Fig. 2a). Sub-clustering of the 7,270 tumor-derived epithelial cells has revealed 18 distinct cell clusters. When the clusters were analyzed based on each patient, most sub-clusters tended to be patient-specific. However, a portion of the epithelial cells derived from

each individual tumor clustered together to form multi-patient clusters (original Supplementary Fig. 2e,f; clusters 1, 10, 11, 16, and 17), suggesting that they are residual non-malignant cells in tumor tissues. The distinction of patient-specific tumor cell clusters from multi-patient normal epithelial cell clusters is a well-known phenomenon in single-cell RNA sequencing (Mario Suva et al., PMID31327527).

Therefore, to clarify the malignancy of tumor tissue-derived epithelial cells, we have used chromosomal gene expression patterns to infer tumor-specific copy number variations (CNVs). Subsequently, tumor-derived epithelial cells without CNV were defined as “non-malignant” cells (918 cells). After excluding the “non-malignant” cells, unsupervised clustering of 10,055 epithelial cells (7270-918=6,352 malignant cells from tumors and 3,703 normal epithelial cells) generated clusters that were mainly distinguished by the tissue origin, tumor, and normal regions (original Supplementary Fig. 2i).

As two reviewers pointed out that the original presentation of cancer cell analysis was extremely confusing, we simplified the section as explained in the answer to the comment #1 (depicted in the Reviewer's ONLY Fig. 1).

10. The authors re-classified DCs into 6 subsets (line 216) as indicated by Villani et al. However, the six identified DC subtypes in that Science paper were collected from PBMC and some of these subtypes have been proven to be not DCs (Dutertre et al, Immunity, 2019). Perhaps this should be re-evaluated?

Author's Response: We appreciate this valuable information. We have checked the above mentioned reference and revised our cell-type assignment for DC subtypes using Villani's [PMID28428369] and Dutertre's [PMID31474513] annotations. As described in the response to question #7, based on the clustering results and marker gene expression, our DCs in clusters #4/#14/#12 were clearly distinguished from monocytes in cluster #3 (Fig. 4a,b). In the Dutertre's study [PMID31474513], CD1c^{lo}CD14⁺ cells have been suggested as one of the conventional DC2 (cDC2) subsets, which were distinguished from monocytes and formed an independent cluster. We have found that the CD163+CD14+ cDC2s identified in Dutertre' is similar to the “CD14+ preDCs” (Fig. 4d-i), which showed an expression of the referred marker genes (cDC2: *LMNA*, *CDKN1A*, *F13A1*, *FCER1A*; CD163+CD14+ cDC2: *S100A8*, *S100A9*, *CD14*). Accordingly, we have changed the annotation to CD163+CD14+ DCs.

11. The authors identified 14 B cell clusters (line 230) but did not annotate these subtypes. Are these biologically meaningful clusters or only technical products driven by batch effects or algorithms? Additional discussion of these subtypes should be included.

Author's Response: The 14 clusters were observed to slightly differ in transcriptional states; however, they were observed to share some gene expression patterns, and were hence, re-grouped. The regrouping was performed based on the overlap between the cluster DEGs (Jaccard index), and presented as the revised Supplementary Fig. 7b.

Revised Supplementary Fig. 7b

Regrouping of B cell clusters using Jaccard index between cluster DEGs

Minor points:

1. This manuscript should be edited by native speakers as there are many language mistakes.

Author's Response: As per the reviewer's suggestion, the manuscript has been revised and edited by native English speakers.

2. The conclusion in line 87 is not precise: "these cellular compositions reflected original tissue microenvironments and gross alterations inflicted by tumor growth and invasion". Single-cell RNA sequencing could only obtain a snapshot and has varied preferences for capturing different cell types, thus it cannot reflect original tissue microenvironment.

Author's Response: We agree with the reviewer's concern that there is a bias in the estimation of cellular compositions obtained by enzymatic dissociation and sequencing. Despite the bias, application of the same tissue dissociation protocol and comparisons of two groups (normal versus tumor) can provide substantial information on the differences and alterations in the tissue microenvironment. We have modified this sentence in the manuscript to present this claim (Lines 103-104) as follows:

"These cellular compositions demonstrated differences in tissue-specific resident populations, as well as gross alterations inflicted by tumor growth and invasion."

3. How the cells are annotated should be clearly documented

Author's Response: The cell types have been annotated based on unsupervised clustering and comparisons of cluster DEGs with known cell type references. We have generated an additional Fig. 1d to depict the global cell type marker expression. We have also used other approaches based on the cell-to-cell correlation analysis for gene expression, which yielded results that were concordant with those of the clustering-based methods (revised Supplementary Fig. 1b). All the reference genes and supporting publications used in the present study are provided in the revised Supplementary Table 2.

4. Detailed calculation of CNV correlation and relevant procedures for identifying malignant cells should be clearly described in Methods.

Author's Response: The method was used as described in a previous study by Chung W et al (PMID28474673) and Puram SV et al. (PMID29198524). We have revised the description in the methods section and provided the computational code at the Github repository (<https://github.com/SGL-LungCancer/SingleCell>).

5. The order of some of the Extended Data Figs. described in the main text is off. For example: Lines

102 and 105: the description of Extended Data Fig. 2h appears before Extended Data Fig. 2g.; Line 138: Extended Data Fig. 3e -> Extended Data Fig. 3b

Author's Response: These corrections have been made.

6. The description in line 107 is confusing: "We performed unsupervised trajectory analysis (Fig. 2a) after excluding non-malignant cells". Why did these non-malignant appear in Fig. 2a after their removal?

Author's Response: The trajectory shown in the original Fig. 2a contains normal lung epithelial cells (Alveolar Types I (AT1) and II (AT2), club cells, ciliated cells, and undetermined cells) and malignant cells. The "malignant cells" refer to epithelial cells isolated from primary lung tumors (tLung) harboring perturbations of inferred CNV signals. The removed non-malignant cells refer to the CNV-negative cells from tLung.

We acknowledge that this confusion arose from the complex analysis scheme, as well as our dual usage of the term "non-malignant cells" for the normal tissue-derived epithelial cells and CNV-negative epithelial cells isolated from tumor tissues. To avoid confusion, we have simplified the analysis scheme (explained in the response to #1) and revised the manuscript accordingly.

Reviewer #2 (Remarks to the Author):

In their paper, Kim et al. present a detailed study of the cellular environment of lung adenocarcinoma, using single cell RNA-seq (Chromium 10X 3'end tag sequencing). In contrast to previous single cell lung cancer profiling papers, the authors of this manuscript concentrate on the changes that occur in metastatic disease, which is of course responsible for most mortality associated with lung cancer.

They generate a large data set of around 250,000 cells that is very heavily immune enriched. The authors carry out detailed computational analysis of the different cell populations that were retrieved, namely epithelial cells, fibroblasts and endothelial cells and different immune cell lineages. For the cancerous epithelial cells the authors identify three distinct cancer states and derive gene signatures for each of these and show that tumor signature 2 is associated with reduced patient survival.

They examine endothelial and fibroblast states and identify distinct sub-populations that differ in their abundance depending on the origin of the cells. (e.g normal, versus tumour, versus LN from metastatic cancer). They further home into the abundance of specific myeloid and macrophage cell states, as well as the distribution of distinct B and T lymphocytes and NK cells.

Overall, they have reasonable representation of epithelial cells, but only relatively small contributions of endothelial and fibroblasts. As expected only the lung tissue itself contributes to fibroblasts and endothelial cells. A very large proportion of the cells they retrieve are immune cells. However, immune enrichment is seen for many dissociation protocols, so this is not unusual.

Lastly, they examine the tissue distribution of immune and stromal cells. They find that exhausted CD8+ T/mo-Mac cells are enriched in metastasis associated tissue. The authors also examine possible receptor-ligand interactions between different cell populations and find that interactions of tS2 signature malignant cells appear to be particularly high with myeloid cells and suggest particular receptor-ligand pairs that may be relevant.

This is a very comprehensive study that is of great interest for better understanding the interplay between the immune system and developing and evolving cancer cells and I believe this study will be of great interest to many readers.

However, there are a number of points the author should address to clarify their analysis or place their results more clearly into the context of the existing literature.

1. As already alluded to the cell purification method appear to enrich for immune cells. In this particular study this is of course very welcome. However, the work would benefit from also having some data that highlights what the actual cell type distribution is. This could be done by cell staining of tissue sections or by single nuclear sequencing of some remaining tumour section. If no more tumour is available the authors should at least highlight that the described cell type distribution is unlikely to reflect the "real" cell type distribution.

Author's Response: We also acknowledge the bias in the estimation of cellular compositions obtained through enzymatic dissociation and sequencing. An over-representation of immune cells has been demonstrated in the previous literature by comparing the bulk transcriptome-inference data and single-cell RNA sequencing data. We do not have left over materials to perform whole-transcriptome sequencing or immunohistochemistry in the matched samples.

As an alternative, we have obtained tumor purity data for the primary resected tumors from the clinic and presented the comparison with the data obtained from the single-cell RNA sequencing (revised Supplementary Fig. 2b). The results have demonstrated a poor correlation between the two sets of data and the influence of the histological type of LUAD in the tumor recovery rate in the single-cell RNA sequencing analysis. We have revised the relevant information in the manuscript to clearly state the biases. In the original manuscript, the information was presented as follows (Lines 73-78):

"Cancer cell recovery rate reflected the physical nature of the histological type and structural conformation of each cell type. Thus, we estimated immune and stromal cell proportions within non-

epithelial compartments after removing the proportional imbalance of epithelial and cancer cells among samples.”

In the revised manuscript, the information has been presented as follows (Lines 87-92):

“Due to the bias introduced during tissue dissociations⁹, single-cell RNA sequencing data overestimated the immune cell proportions in comparison to the stromal and epithelial cell types (Supplementary Fig. 2a). In addition, the recovery rate of tumor cells was affected by the histological types of LUAD (Supplementary Fig. 2b). Therefore, we assessed the compositions of immune cell subsets after removing the epithelial and stromal populations.”

2. Figure 2a: The authors say: “We performed unsupervised trajectory analysis (Fig. 2a) after excluding non-malignant cells to adjust for inter-patient genomic heterogeneity and to find key cancer gene expression programs (Extended Data Fig. 2i).”

However, the trajectory that is shown in Fig 2a clearly contains malignant and non-malignant cells. The author must explain this more clearly.

Author’s Response: The trajectory shown in Fig. 2a contains normal tissue-derived lung epithelial cells (Alveolar Types I (AT1) and II (AT2), club cells, ciliated cells, and undetermined cells) and malignant cells. The “malignant cells” were derived from primary lung tumors (tLung) that harbored perturbation of inferred CNV signals. The removed “non-malignant cells” refer to the CNV-negative epithelial cells isolated from tumor tissues.

In the original submission, we aimed to provide an extensive view of the intrinsic characteristics of adenocarcinoma cells reflecting inter-patient heterogeneity and differentiation programs. In the complex scheme, we performed comparative analysis between normal and tumor epithelial cells before and after: 1) adjusting for batch correction algorithms, 2) excluding non-malignant cells present in cancer tissues.

In the revision, we have simplified the section on cancer cell analysis (Reviewer’s ONLY figure 1 in the response to the Reviewer#1) and focused on the main point, the tS2 tumor signature.

3 The authors derive three tumour cell gene expression signatures according to the three branches of the pseudo time trajectory. They go on to say that gene signature S2 is associated with poor outcome in LUAD. I feel the authors need to discuss how this relates to previously defined LUAD cancer gene signatures. Does this gene signature overlap with those previously described? How well does S2 correlate with poor outcome? Would a classic proliferation signature do better?

Suppl Fig 3 is meant to explain this further. However, there seems to be no colour code for Fig S3a. What are the colours in the first tSNE? It is labelled with tS1, tS2 and tS3, but the distribution of these clusters is very different from tS1-3 as colour coded in the tSNE for cell states. The authors must clarify this.

Author’s Response: Gene expression subtypes of lung adenocarcinoma (LUAD) have been reported in various studies using bulk RNA sequencing and mRNA microarray data (The Cancer Genome Atlas Research Network’s, PMID 25079552; Beer et al, (PMID12118244); and Bhattacharjee et al, (PMID11707567)). In a previous study (PMID25079552), the TCGA samples have been defined as three subtypes using 230 LUAD samples, including the terminal respiratory unit (TRU, formerly bronchioid), the proximal-inflammatory (PI, formerly squamoid), and the proximal-proliferative (PP, formerly magnoid), which was previously suggested in studies by Hayes DN [PMID17075127] and Wilkerson MD [PMID2259055].

To compare the survival outcome with that of the present study, we have analyzed the Kaplan-Meier curves using the representative genes for each subtype (presented in the Supplementary figure 7B of PMID25079552) on 494 TCGA LUAD samples used in our study (Reviewer’s ONLY Figure 3). The

tumor samples were divided into two classes (low and high) as the 25th and 75th percentiles of the mean expression of the target genes. The patterns of overall survival for the high group were observed to be same as those reported previously: TRU > PP > PI. However, the difference of outcomes between the high and low group was only significant for TRU. The representative signature genes for TRU included *SFTPC*, *DMBT1*, and *FOLR1*. Among these, *SFTPC* was observed to be the top signature for our tS1 tumor cell state. The signature genes for PP and PI did not overlap with our tS2- or tS3-specific signatures; however, their expression showed weak upregulation in tS2.

Reviewer's ONLY Figure 3. Kaplan-Meier overall survival curves based on gene signatures of previously defined LUAD subtypes [PMID25079552] in our validation TCGA LUAD cohorts. +: censored observations. P-value (p) was calculated using the two-sided log-rank test.

Author's response on confusing data presentation in Supplementary Fig. 3:

In original manuscript, we overlaid the labels representing clusters for each color on the first tSNE plot in Supplementary Fig. 3a. We have consistently confirmed the tS2-enriched clusters in the tLung (tLung-Cancer-CCA-C0 and 4 in Supplementary Fig. 3a) and all tumor samples (All-Cancer-CCA-C2 and 3 in Supplementary Fig. 3c). As in the response to question #2, we have simplified the section on cancer cell analysis (Reviewer's ONLY figure 1 in the response to the Reviewer#1).

4. The authors state: "The efficacy of targeted agents against EGFR signaling demonstrated minimal association with intra- and inter-tumoral heterogeneity" but must make clear that is predicted rather than actual efficacy.

Author's Response: We appreciate the reviewer's comment. As explained in the response #2, we have simplified the cancer cell analysis to focus on the tS2 signature, and the drug response prediction was removed from the manuscript as it was not our main focus.

5. Figure 3d: It is not clear exactly what this figure is depicting. There is very little explanation. "The most significantly upregulated genes in tumor ECs were associated with angiogenesis, and strong activation of VEGFR and Notch signaling (Fig. 3b,d, Supplementary Table 4)." Could the authors include some explanation of how their functional association network was derived.

Author's Response: We have revised the manuscript to provide more information regarding the generation of figures, including the section on the endothelial cells. Nodes on the network represent the upregulated and downregulated genes specific to tumor ECs, and edges represent the association score defined as the sharing ratio of the biological processes of gene ontology (GO) between two genes. In the network, tumor EC-specific genes were clustered on the basis of their annotated GO terms (Fig. 3d, Supplementary Table 5). As a result, we have observed the discriminative functional category "angiogenesis" assigned to the upregulated gene group. Taken

together, genes in VEGFR and Notch signaling were over-expressed in tumor ECs (Fig. 3b). We have revised this information in the manuscript as follows (Lines 170-174):

“Tumor ECs demonstrated a strong activation of VEGF and Notch signaling (Fig. 3b), which regulates the development and cell fate determination of endothelial cells^{18, 19}. Gene expression network analysis of tumor ECs further highlighted “angiogenesis” as the upregulated genes’ functional category (Fig. 3d, Supplementary Table 5).”

In the revision, we have added the ligand-receptor interaction data in the revised Fig. 7a and b. Most ligand-receptor pairs between tS2/Malignant cells and tumor ECs were associated with angiogenesis signaling molecules, such as VEGF-VEGFRs and ephrin-Eph receptors (PMID22866201). As expected, tumor ECs received angiogenic stimulatory signals from mo-Macs/malignant cells for VEGF and its receptor *FLT1/VEGFR1*, *KDR/VEGFR2*, as a key mediator of angiogenesis in cancer (PMID16301830, PMID16336962, PMID24314323) for samples of all tumor stages or for brain metastases (Lines 322-327).

6. There are extensive tables with gene signatures, but I think it would be helpful to have a supplementary table, but ideally figure that indicates the marker genes that were used to annotate each cell type. (apologies if I have overlooked such a figure.). This will make it much easier to other groups to relate their work to this study.

Author’s Response: As the reviewer has pointed out, it is important to employ unique and reliable gene signatures for the accurate cell type assignment. Therefore, we have presented the expression map of individual markers in Supplementary Fig. 1a, Supplementary Fig. 2b, Fig. 3b and g, Fig. 4b and e, Supplementary Fig. 6a, and Supplementary Fig. 7a in the original submission.

In the revision, we additionally provide an expression map of individual markers for the nine cell lineages in the revised Fig. 1d. List of reference genes are provided in the revised Supplementary Table 2.

Revised Fig. 1d

7. minor point: in Figure 5b exhausted is mis-spelt.

Author’s Response: This correction has been made.

The computational packages used in the analysis employ a wide range of statistical tests. I am afraid I am not able to comment on the suitability and correct application of these tests.

Author's Response: Per the reviewer's suggestion, we have included the results of two-sided Student's t-test, two-sided Wilcoxon test, and one-way ANOVA test in an appropriate format throughout the revised manuscript (t-test, Fig. 3j, Fig. 4h and k, Fig. 6b and c, Supplementary Fig. 2d, Supplementary Fig. 5c and e, Supplementary Fig. 6b, Supplementary Fig. 7d, Supplementary Fig. 8c; Wilcoxon test, Fig. 2e; ANOVA test, Fig. 2f, Fig. 4i, Fig. 5f). This will complement other statistical methods we have used, which can be easily understood by a broad readership.

REVIEWERS' COMMENTS:

Reviewer #1 (Remarks to the Author):

In general, I'm satisfied by the response and the revision. However, I have a few points that need to be addressed:

1. In line 129-131, the authors claimed that "the separation of tS2 from the normal epithelial cells were repeatedly observed". However, in the individual trajectory analysis, tS1 cells were also separated from the normal cells, differing from the results in the integrated analysis.

2. The authors need to make Figure 2A easier to read. Malignant cells in branch S3 were almost indistinguishable from the ciliated cells.

3. In Figure 4H, does y coordinate really represents Ro/e ? I'm confused about the order of magnitude.

4. A sentence in Figure Legend 4 "(RO/E is) the ratio of observed cell numbers to random expectation calculated by chi-square test" is exactly same as the one in the paper PMID29942094 provided by the authors. The authors should rewrite this sentence and check the manuscript throughout to avoid such irregularities.

Reviewer #2 (Remarks to the Author):

The authors have made substantial changes to the manuscript and have altered a number of the analyses that were confusing in the first version of the paper. I believe the paper is now much clearer and the concerns I raised in my first review have been addressed.

I was pleased to see that a web portal for browsing the data has been created. However, despite trying the password I was unable to use the browser. The authors must make sure that this site is accessible upon publication.

Point-by-point response to the Reviewer's comments

REVIEWERS' COMMENTS:

Reviewer #1 (Remarks to the Author):

In general, I'm satisfied by the response and the revision. However, I have a few points that need to be addressed:

1. In line 129-131, the authors claimed that "the separation of tS2 from the normal epithelial cells were repeatedly observed". However, in the individual trajectory analysis, tS1 cells were also separated from the normal cells, differing from the results in the integrated analysis.

Author's Responses: As individual tumors may not contain all three tumor molecular subtype, the trajectory data structure is better represented by the aggregated analysis. For example, P0034 and P0028 patients did not have tS3 cells and P0028 did not have normal ciliated cells. Thus individual trajectory did not form a perfectly identical branch structure with the integrated analysis. Nonetheless, we repeatedly observed the separation of tS2 from the normal epithelial cells. In response to the reviewer's criticism, we added an explanation for the discrepancy.

We have revised the sentence "In the individual patient-by-patient trajectories, the separation of tS2 from the normal epithelial cells was repeatedly observed (Supplementary Fig. 3e)." as "In the individual patient-by-patient trajectories, the separation of tS2 from the normal epithelial cells was repeatedly observed (Supplementary Fig. 3e) despite the different trajectory structure in each patient due to the limited representation of cellular components." in the revised manuscript.

Supplementary Fig. 3e

2. The authors need to make Figure 2A easier to read. Malignant cells in branch S3 were almost indistinguishable from the ciliated cells.

Author's Responses: This is due to the extensive overlap of the dots (=cells). We inserted a panel with an each cell type highlighted at the bottom.

Fig. 2a

3. In Figure 4H, does y coordinate really represents Ro/e? I'm confused about the order of magnitude.

Author's Responses: In Fig. 4h, Ro/e was calculated in a combination of samples and myeloid cells using `chisq.test` function in R. In the cases of cell subsets which have a biased observed value in a specific sample, those Ro/e showed extreme value.

4. A sentence in Figure Legend 4 "(RO/E is) the ratio of observed cell numbers to random expectation calculated by chi-square test" is exactly same as the one in the paper PMID29942094 provided by the authors. The authors should rewrite this sentence and check the manuscript throughout to avoid such irregularities.

Author's Responses: That is the original definition of Ro/e. We have revised the sentence as "Ro/e is the relative score of observed cell numbers over expected cell numbers calculated by chi-square test".

Reviewer #2 (Remarks to the Author):

The authors have made substantial changes to the manuscript and have altered a number of the analyses that were confusing in the first version of the paper. I believe the paper is now much clearer and the concerns I raised in my first review have been addressed.

I was pleased to see that a web portal for browsing the data has been created. However, despite trying the password I was unable to use the browser. The authors must make sure that this site is accessible upon publication.

Author's Responses: The password is now removed. We updated and widely tested our web-based site (<http://ureca-singlecell.kr>) for stable operation.